# Universal Functional Regression with Neural Operator Flows

**Yaozhong Shi**  *yshi5@caltech.edu*
*California Institute of Technology*

**Angela F. Gao**  *afgao@caltech.edu*
*California Institute of Technology*

**Zachary E. Ross**  *zross@caltech.edu*
*California Institute of Technology*

**Kamyar Azizzadenesheli**  *kamyara@nvidia.com*
*NVIDIA Corporation*

**Reviewed on OpenReview:** *https://openreview.net/forum?id=rHL329Xa3X*

## Abstract

Regression on function spaces is typically limited to models with Gaussian process priors. We introduce the notion of universal functional regression, in which we aim to learn a prior distribution over non-Gaussian function spaces that remains mathematically tractable for functional regression. To do this, we develop Neural Operator Flows (OPFLOW), an infinite-dimensional extension of normalizing flows. OPFLOW is an invertible operator that maps the (potentially unknown) data function space into a Gaussian process, allowing for exact likelihood estimation of functional point evaluations. OPFLOW enables robust and accurate uncertainty quantification via drawing posterior samples of the Gaussian process and subsequently mapping them into the data function space. We empirically study the performance of OPFLOW on regression and generation tasks with data generated from Gaussian processes with known posterior forms and non-Gaussian processes, as well as real-world earthquake seismograms with an unknown closed-form distribution.

## 1 Introduction

The notion of inference on function spaces is essential to the physical sciences and engineering. In particular, it is often desirable to infer the values of a function everywhere in a physical domain given a sparse number of observation points. There are numerous types of problems in which functional regression plays an important role, such as inverse problems, time series forecasting, data imputation/assimilation. Functional regression problems can be particularly challenging for real world datasets because the underlying stochastic process is often unknown.

Much of the work on functional regression and inference has relied on Gaussian processes (GPs) (Rasmussen & Williams, 2006), a specific type of stochastic process in which any finite collection of points has a multivariate Gaussian distribution. Some of the earliest applications focused on analyzing geological data, such as the locations of valuable ore deposits, to identify where new deposits might be found (Chiles & Delfiner, 2012). GP regression (GPR) provides several advantages for functional inference including robustness and mathematical tractability for various problems. This has led to the use of GPR in an assortment of scientific and engineering fields, where precision and reliability in predictions and inferences can significantly impact outcomes (Deringer et al., 2021; Aigrain & Foreman-Mackey, 2023). Despite widespread adoption, the assumption of a GP prior for functional inference problems can be rather limiting, particularly in scenarios where the data exhibit heavy-tailed or multimodal distributions, e.g. financial time series or environmental modeling. This

underscores the need for models with greater expressiveness, allowing for regression on general function spaces, with data arising from potentially unknown stochastic processes and distributions (Fig. 1). We refer to such a regression problem as universal functional regression (UFR).

We hypothesize that addressing the problem of UFR requires a model with two primary components. First, the model needs to be capable of learning priors over data that live in function spaces. Indeed, there has been much recent progress in machine learning on learning priors on function spaces (Rahman et al., 2022a; Lim et al., 2023; Seidman et al., 2023; Kerrigan et al., 2023a; Pidstrigach et al., 2023; Baldassari et al., 2023; Hagemann et al., 2023; Kerrigan et al., 2023b; Franzese et al., 2023). Second, the models need a framework for performing functional regression with the learned function space priors and learned likelihood–a component that is critically missing from the previous works on pure generative models described above. This includes the ability to compute the likelihood of an input data sample, which would allow for posterior estimation over the entire physical domain. Addressing these challenges not only expands the models available for functional regression but also enhances our capacity to extract meaningful insights from complex datasets.

In this study, our contributions are as follows. We develop a viable formulation for UFR with a novel learnable bijective operator, OPFLOW. This operator extends the classical normalizing flow, which maps between finite dimensional vector spaces, to the function space setting. OPFLOW consists of an invertible neural operator, an invertible map between function spaces, trained to map the data point-process distribution to a known and easy-to-sample GP distribution. Given point value samples of a data function, OPFLOW allows for computing the likelihood of the observed values, a principled property a priori held by GP. Using this property, we formally define the problem setting for UFR and develop an approach for posterior estimation with Stochastic Gradient Langevin Dynamics (SGLD) (Welling & Teh, 2011). Finally, we demonstrate UFR with OPFLOW on a suite of experiments that use both Gaussian and non-Gaussian processes.

In comparison to GPR, OPFLOW draws on the training dataset to learn the prior over a function space. When such datasets are only minimally available, kernel tuning methods propose to hand-tune the kernel and parameters of GPR, providing considerable performance, although still with a Gaussian assumption. In such limited data scenarios, learning accurate priors with OPFLOW may be difficult and less suitable. When training data is non-existent, often neither the plain GPR nor OPFLOW can be of help; when expert knowledge can be incorporated (Aigrain & Foreman-Mackey, 2023), there may be advantages to GPR. However, such a method can also be considered as hand-tuning the prior used in OPFLOW by, for example, setting the invertible model to identity map, resembling strong prior over the prior learning process.

## 2 Related work

**Neural Operators.** Neural Operator is a new paradigm in deep learning for learning maps between infinite-dimensional function spaces. Unlike traditional neural networks that primarily work with fixed-dimensional vectors, neural operators are designed to operate on functions, making them inherently suitable for a wide range of scientific computing and engineering tasks involving partial differential equations (PDEs) (Kovachki et al., 2023; Li et al., 2020). A key feature of neural operators is their discretization agnostic (resolution invariant) property (Kovachki et al., 2023), which means that a neural operator can learn from data represented on various resolutions and predict outcomes on yet new resolutions. This property is particularly valuable in applications involving the natural sciences, PDEs, and complex physical simulations where functional data may come from variable discretizations of meshes or sensors. Within the family of neural operators, Fourier Neural Operator (FNO) (Li et al., 2021) stands out for its quasi-linear time complexity by defining the integral kernel in Fourier space and has been applied to many problems in engineering and science (Azizzadenesheli et al., 2024). In the conventional view of neural operators, provided discretization and point evaluation of functions resembles approximation of function for which increasing the resolution is connected to improved approximation error in the integration and differentiation operators (Liu-Schiaffini et al., 2024). The proposed view in the current work provides a new perspective for which the point evaluations are instead considered as points samples of functions that are associated

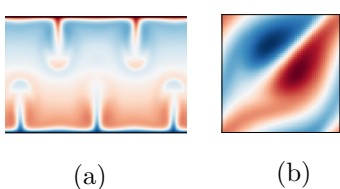

(a)                    (b)

Figure 1: Examples of data from non-Gaussian processes. (a) Temperature field from Rayleigh Bénard convection problem. (b) Vorticity field from Naiver-Stokes Equation.

probably rather than finite bases and finite resolution approximation of a continuous function. Therefore, increasing the resolution is seen as providing more information for regression rather than a way to reduce, e.g., the approximation error in the integral operators used in neural operations.

**Function space generative modeling with Neural Operators.** Several discretization invariant generative models on function spaces have been recently developed using Neural Operators (Li et al., 2020; Kovachki et al., 2023). These models include Generative Adversarial Neural Operator (GANO) (Rahman et al., 2022a), Denoising Diffusion models (DDO) (Lim et al., 2023) and Variational Autoencoding Neural Operator (VANO) (Seidman et al., 2023). To be specific, GANO is the first proposed infinite-dimensional generative model that generalizes GAN, and learns probability measures on function spaces. However, GANO training relies on an adversarial game, which can be slow (Goodfellow et al., 2014; Rahman et al., 2022a); DDO is a resolution-invariant diffusion generative model for function spaces, leveraging denoising score matching within Hilbert spaces and employing infinite-dimensional Langevin dynamics for sampling. The inference of DDO requires a diffusion process, as with the finite-dimensional score-based model, which can be time-consuming (Song et al., 2021; Lim et al., 2023); VANO offers a functional variational objective (Kingma & Welling, 2022; Seidman et al., 2023), mapping input functional data to a finite-dimensional latent space through an encoder, which may be a sub-optimal formulation due to the fixed dimensionality of the latent space being decoupled from the discretization of the input function data.

All of these aforementioned models have exhibited superior performance over their finite-dimensional counterparts for functional data by directly learning mappings between function spaces, which show great potential in addressing the challenges in science and engineering problems (Shi et al., 2024a;b; Yang et al., 2021; Azizzadenesheli et al., 2024). Compared to these existing infinite-dimensional generative models, OPFLOW shows advantages in fast training, fast inference, bijective architecture, and providing precise likelihood estimation as summarized in Table 1. These advantages of OPFLOW can be explained by: (i) The training and inference processes of OPFLOW only involve likelihood estimation, and don't require an adversarial game or a diffusion process. (ii) OPFLOW directly generalizes normalizing flows to function space without finite-dimensional simplicity, and enables precise likelihood estimation for functional point evaluations on irregular grids. (iii) As an invertible operator, OPFLOW can be useful for tasks that require the bijectivity between function spaces (Zhou et al., 2023; Furuya et al., 2023).

Table 1: Comparison of OPFLOW with other infinite-dimensional generative models

| Models | Fast training | Fast inference | Bijective architecture | Exact likelihood estimation |
|---|---|---|---|---|
| GANO | ✗ | ✓ | ✗ | ✗ |
| DDO | ✗ | ✗ | ✗ | ✗ |
| VANO | ✓ | ✓ | ✗ | ✗ |
| OPFLOW [Ours] | ✓ | ✓ | ✓ | ✓ |

**Neural Processes.** Neural Process (NP) (Garnelo et al., 2018) leverages neural networks to address the constraints associated with GPs, particularly the computational demands and the restrictive Gaussian assumptions for priors and posteriors (Dutordoir et al., 2022). While NP aims to model functional distributions, several fundamental flaws suggest NP might not be a good candidate for learning function data (Rahman et al., 2022a; Dupont et al., 2022). First, NP lacks true generative capability; it focuses on maximizing the likelihood of observed data points through an amortized inference approach, neglecting the metric space of the data. This approach can misinterpret functions sampled at different resolutions as distinct functions, undermining the NP's ability to learn from diverse function representations. As pointed out in prior works (Rahman et al., 2022a), if a dataset comprises mainly low-resolution functions alongside a single function with a somewhat high resolution, NP only tries to maximize the likelihood for the point evaluation of the high-resolution function during training and ignores information from all other lower-resolution functions. A detailed experiment to show the failure of NP for learning simple function data can be found in Appendix A.1 of (Rahman et al., 2022a) . What's more, NP relies on an encoder to map the input data pairs to finite-dimensional latent variables, which are assumed to have Gaussian distributions (Garnelo et al., 2018), and then project the finite-dimensional vector to an infinite-dimensional space; this approach results in diminished consistency at elevated resolutions, and thus disables NP to scale to large datasets (Dupont et al.,

2022). The other fundamental limitation is the Bayesian framework of NP is also defined on a set of points, rather than functions themselves. This method results in the dilution of prior information as the number of evaluated points increases, further constraining NP's ability to generalize from function data.

**Normalizing Flows.** Normalizing flows are a class of flow-based finite-dimensional generative models, that usually composed of a sequence of invertible transformations (Kingma & Dhariwal, 2018; Dinh et al., 2017; Chen et al., 2019). By gradually transforming a simple probability distribution into a more complex target distribution with the invertible architecture, normalizing flows enable exact likelihood evaluation, direct sampling, and good interpretability (Kobyzev et al., 2021) with applications from image and audio generations (Kingma & Dhariwal, 2018; Ping et al., 2020), reinforcement learning (Mazoure et al., 2020) to Bayesian inference (Gao et al., 2021; Whang et al., 2021). Despite the advantages of normalizing flows, they still face challenges with high-dimensional data due to computational complexity, memory demands, scalability, and cannot be directly used for zero-shot super-resolution (Lugmayr et al., 2020). Moreover, traditional normalizing flows are defined on a finite-dimensional spaces, and constrained to evaluating likelihood on fixed-size, regular grids. This inherent limits their applications in Bayesian inverse problems, where the data may not conform to fixed and regular grids (Dashti & Stuart, 2017). These limitations underline the need for a new generation of normalizing flows designed on function space, providing a solution that could potentially transcend the limitations of traditional models.

**Nonparametric Bayesian and Gaussian Process Regression.** Nonparametric Bayesian models are defined on an infinite-dimensional parameter space, yet they can be evaluated using only a finite subset of parameter dimensions to explain observed data samples (Orbanz & Teh, 2010). These models provide a flexible regression framework by offering analytically tractable posteriors and enable adjusting their complexity by allowing the model size to grow when more observations are available. Further, nonparametric Bayesian leverages a rich theoretical foundation of parametric Bayesian modeling and can be evaluated using common inference techniques like Markov Chain Monte Carlo (MCMC) and variational inference, etc. Despite the great advantages of the nonparametric Bayesian models, such models heavily depend on the chosen prior, which could restrict their adaptability and scalability in real-world complex scenarios (Barry, 1986; Wasserman, 2006). The emergence of learning-based Bayesian models, propelled by advancements in deep learning, offers a potential solution by marrying the flexibility and scalability of neural networks with Bayesian principles; these modern approaches learn directly from data rather than rely on the chosen priors.

Within the family of nonparmateric Bayesian models, GPR is distinguished for its widespread applications, characterized by its analytical posteriors within the Bayesian framework (Rasmussen & Williams, 2006). GPR offers a robust framework for capturing uncertainties in predictions given noisy observations. However, GPR assumes data can be modeled using a Gaussian process, both the prior and posterior should be Gaussian. Such assumption, while simplifying the analytical process, might not always align with the complex nature, thus the universal applicability of GPR is constrained by the diversity and complexity of real-world datasets.

**Deep and Warped Gaussian Processes.** Deep GP stacks multiple layers of Gaussian process (Damianou & Lawrence, 2013), which enables modeling of highly nonlinear and potentially non-Gaussian data, as well as providing uncertainty quantification at each layer (Jakkala, 2021; Liu et al., 2020; de Souza et al., 2024). However, as Ustyuzhaninov et al. (2020); Havasi et al. (2018) have pointed out, achieving exact Bayesian inference in Deep GP is challenged by its inherent intractability, necessitating the use of simplifying approximations. Unfortunately, these approximations could lead to the layers of a Deep GP collapsing to near-deterministic transformations (Ustyuzhaninov et al., 2020), significantly reducing the model's probabilistic nature and its ability to quantify uncertainty accurately. Separately, Warped GP presents an alternative for modeling non-Gaussian data by either transforming the likelihood of the outputs of GP with nonlinear transformations (Snelson et al., 2003) or transforming the prior. While Warped GP has shown efficacy in handling real-world non-Gaussian data (Kou et al., 2013), the selection of an appropriate warping function is critical and demands extensive domain expertise, where an ill-suited warping function can severely degrade model performance (Rios & Tobar, 2019). Besides, similar to Deep GP, Warped GP also suffers from analytical intractability, often requiring the employment of variational inference methods for practical implementation (Lázaro-Gredilla, 2012). Recently, inspired by the success of normalizing flows, (Maroñas et al., 2021) proposed Transformed GP, which utilizes a marginal normalizing flow to learn the non-Gaussian prior. This approach not only matches the capability of traditional Deep GP in modeling complex data,

but is also computationally efficient. However, Transformed GP use, yet again, pointwise wrapping of GP point evaluation, resulting in marginal flows that perform diagonal mapping, limiting its applicability to learn more general stochastic process. Additionally, the inference of Transformed GP relies on sparse variational objective which is considered as a sub-optimal approach in terms of accuracy.

Regression with OPFLOW addresses the limitations of existing Gaussian process models discussed above, and offer two key improvements over the Transformed GP framework (Maroñas et al., 2021). To be specific, the OPFLOW enables modeling a more general stochastic process compared to the marginal flow which is based on pointwise operator. The regression with the OPFLOW enables true Bayesian uncertainty quantification via drawing posterior samples of the Gaussian process with SGLD and subsequently mapping them into the data function space. Such regression formulation is valid in the sense of correctness, imposing no approximation in the formulation, which enables us to capture true Bayesian uncertainty. The mentioned conceptual advancements are empirically shown beneficial in this paper. In contrast, variational inference formulation often relies on optimizing the Evidence Lower Bound (ELBO) and yields an approximate posterior with an approximation gap "in the formulation", rather than the true posterior. Despite the benefit, the proposed regression framework with SGLD requires significantly more time for sampling compared to variational inference, where the latter is more practical for time-sensitive application. Last, we should note that the regression framework of OPFLOW slightly differs from that of the generic Deep GP. In Deep GP or Transformed GP, the model is fitted to the entire training dataset. For example, when analyzing a dataset containing rainfall measurements over a ten-year period, a Deep GP or Transformed GP would be used to directly fit all the data, enabling predictions of rainfall at any point within these ten years. In contrast, the regression with OPFLOW can be analogous to image inpainting or reconstruction from pixels with generative models in a Bayesian manner (Marinescu et al., 2021). In the OPFLOW framework, each data point in the training set represents a distinct function, potentially having its unique discretization. During training, the goal of OPFLOW is to learn a bijective mapping between a Gaussian process and the data function space. Once trained, OPFLOW will be frozen and serve as the prior, when provided with new observations (i.e., point evaluations of an unknown function drawn from the data function measure), the task is to reconstruct the entire function, aligning OPFLOW closely with the traditional settings of standard GP regression. The regression setup with OPFLOW is formally defined in the subsequent section.

## 3 Preliminaries

**Universal functional regression.** Consider a function space, denoted as $\mathcal{U}$, such that for any $u \in \mathcal{U}, u : \mathcal{D}_{\mathcal{U}} \to \mathbb{R}^{d_{\mathcal{U}}}$. Here, $\mathcal{U}$ represents the space of the data. Let $\mathbb{P}_{\mathcal{U}}$ be a probability measure defined on $\mathcal{U}$. A finite collection of observations $u|_D$ is a set of point evaluations of function $u$ on a few points in its domain $D \subset \mathcal{D}_{\mathcal{U}}$, where $D$ is the collection of co-location positions $\{x_i\}_{i=1}^l, x_i \in \mathcal{D}_{\mathcal{U}}$. Note that in practice, $u \in \mathcal{U}$ is usually not observed in its entirety over $\mathcal{D}_{\mathcal{U}}$, but instead is discretized on a mesh. In UFR, independent and identically distributed samples of $\{u_j\}_{j=1}^m \sim \mathbb{P}_{\mathcal{U}}$ are accessed on the discretization points $\{D_j\}_{j=1}^m$, constituting a dataset of $\{u_j|_{D_j}\}_{j=1}^m$. The point evaluation of the function draws, i.e., $u \in \mathcal{U}$, can also be interpreted as a stochastic process where the process is discretely observed. The behavior of the stochastic process is characterized by the finite-dimensional distributions, such that for any discretization $D^\dagger$, $p(u|_{D^\dagger})$ denotes the probability distribution of points evaluated on a collection $D^\dagger$. What's more, $p(u|u|_{D^\dagger})$ is $p(u \ given \ u|_{D^\dagger})$, that is the posterior. Similarly, $p(u|_{D'}|u|_{D^\dagger})$ is the posterior on $D'$ collocation points. In the UFR setting, each data sample $u_j$ is provided on a specific collection of domain points $D_j$, and for the sake of simplicity of notation, we consider all the collected domain points to be the same and referred to by $D$.

The main task in UFR is to learn the measure $\mathbb{P}_{\mathcal{U}}$ from the data $\{u_j|_{D_j}\}_{j=1}^m$ such that, at the inference time, for a given $u|_{D^\dagger}$ of a new sample $u$, we aim to find its posterior and the $u$ that maximizes the posterior,

$$\max_{u \in \mathcal{U}} p(u|u|_{D^\dagger}),$$

as well as sampling from the posterior distribution. Please note that, when $\mathbb{P}_{\mathcal{U}}$ is known to be Gaussian, then UFR reduces to GPR.

**Invertible operators on function spaces.** Consider two Polish function spaces, denoted $\mathcal{A}$ and $\mathcal{U}$, such that for any $a \in \mathcal{A}, a : \mathcal{D}_{\mathcal{A}} \to \mathbb{R}^{d_{\mathcal{A}}}$, and for any $u \in \mathcal{U}, u : \mathcal{D}_{\mathcal{U}} \to \mathbb{R}^{d_{\mathcal{U}}}$. Here, $\mathcal{U}$ represents the space of the

data, whereas $\mathcal{A}$ represents the latent space. Let $\mathbb{P}_{\mathcal{U}}$ and $\mathbb{P}_{\mathcal{A}}$ be probability measures defined on $\mathcal{U}$ and $\mathcal{A}$, respectively. In this work, we take samples of functions in $\mathcal{A}$, i.e., $a_j \in \mathcal{A}$ to be drawn from a known GP, and consequently, the evaluation of $a_j|_{D_{\mathcal{A}}}$ is a Gaussian random variable equipt with a Gaussian probability measure. Under these construction, and for a suitable Gaussian process on $\mathcal{A}$, we may establish an invertible (bijective) operator between $\mathcal{A}$ and $\mathcal{U}$[1]. We denote the forward operator as $\mathcal{G} : \mathcal{A} \to \mathcal{U}$, such that for any $a_j$, $\mathcal{G}(a_j) = u_j$, and its corresponding inverse operator $\mathcal{G}^{-1}$ as $\mathcal{F}$, where $\mathcal{F} = \mathcal{G}^{-1}$ and $\mathcal{F} : \mathcal{U} \to \mathcal{A}$, such that for any $u_j$, $\mathcal{F}(u_j) = a_j$. We aim to approximate $\mathcal{F}$ and correspondingly its inverse $\mathcal{G}$ with an invertible learnable model $\mathcal{F}_\theta \approx \mathcal{F}$, and $\mathcal{G}_\theta = \mathcal{F}_\theta^{-1}$, from a finite collection of observations $\{u_j|_D\}_{j=1}^m$. Finally, to have well-defined invertible maps between the two measures, we assume absolute continuity of measures on $\mathcal{A}$ and $\mathcal{U}$ with respect to each other. Given the GP nature of the measure on $\mathcal{A}$, such continuity is obtained by adding a small amount of GP noise to the measure on $\mathcal{U}$. A similar procedure is envisioned and incorporated by adding Gaussian noise to images in the prior normalizing flow in finite-dimensional spaces (Huang et al., 2020; Shorten & Khoshgoftaar, 2019).

## 4 Neural Operator Flow

### 4.1 Invertible Normalized Architecture

We introduce the Neural Operator Flow (OpFlow), an innovative framework that extends the principle of the finite-dimensional normalizing flow into the realm of infinite-dimensional function spaces. The architecture is shown schematically in Fig. 2. OpFlow retains the invertible structure of normalizing flows, while directly operating on function spaces. It allows for exact likelihood estimation for point estimated functions. OpFlow is composed of a sequence of layers, each containing the following parts:

**Actnorm.** Channel-wise normalization with trainable scale and bias parameters is often effective in facilitating affine transformations in normalizing flow neural network training (Kingma & Dhariwal, 2018). We implement a function space analog by representing the scale and bias as learnable constant-valued functions. Specifically, let $(v')^i$ denote the input function to the $i$th actnorm layer of OpFlow and let $s_\theta^i$ and $b_\theta^i$ represent the learnable constant-valued scale and bias functions, respectively, in this layer. The output of the actnorm layer is then given by $v^i = s_\theta^i \odot (v')^i + b_\theta^i$, where $\odot$ denotes pointwise multiplication in the function space.

**Domain and codomain partitioning.** The physical domain refers to the space in which the data resides, such as the time domain for 1D data or the spatial domain for 2D data. Conversely, the channel domain is defined as the codomain of the function data, such as the 3 channels in RGB images or temperature, velocity vector, pressure, and precipitation in weather forecast (Pathak et al., 2022; Hersbach et al., 2020). In normalizing flows, a bijective structure requires dividing the input domain into two segments (Dinh et al., 2017; Kingma & Dhariwal, 2018). Based on the method of splitting the data domain, we propose two distinct OpFlow architectures.

*(i) Domain partitioning*: In this architecture, we apply checkerboard masks to the physical domain, a technique widely used in existing normalizing flow models (Dinh et al., 2017; Kingma & Dhariwal, 2018) which we extend to continuous domains. Our experiments in the following sections reveal that the domain decomposed OpFlow is efficient and expressive with a minor downside in introducing checkerboard pattern artifacts in zero-shot super-resolution tasks.

*(ii) Codomain partitioning*: This approach partitions the data along the codomain of the input function data, while maintaining the integrity of the entire physical domain. We show that, unlike its domain decomposed counterpart, the codomain OpFlow does not produce artifacts in zero-shot super-resolution experiments. However, it exhibits a lower level of expressiveness compared to the domain decomposed OpFlow.

**Affine coupling.** Let $v^i$ represent the input function data to the $i$th affine coupling layer of OpFlow. Then, we split $v^i$ either along the physical domain or codomain, and we have two halves $h_1^i, h_2^i = \mathbf{split}(v^i)$. Let $\mathcal{T}_\theta^i$ denote the $i$th affine coupling layer of OpFlow, where $\mathcal{T}_\theta^i$ is a FNO layer (Li et al., 2021), which ensures the model is resolution agnostic. Then, we have $\log(s^i), b^i = \mathcal{T}_\theta^i(h_1^i)$, where $s^i$ and $b^i$ are scale and shift functions, respectively, output by the $i$th affine coupling layer. From here, we update $h_2^i$ through $(h')_2^i = s^i \odot h_2^i + b^i$.

---

[1]We only require homeomorphism between $\mathcal{D}_{\mathcal{A}}$ and $\mathcal{D}_{\mathcal{U}}$ where the point alignment is through the underlying homeomorphic map, and use $D_{\mathcal{A}}$ to represent the discretized domain of $\mathcal{D}_{\mathcal{A}}$. For simplicity, we consider the case where $\mathcal{D}_{\mathcal{U}} = \mathcal{D}_{\mathcal{A}}$ and the trivial identity map is the homeomorphic map in our experimental study.

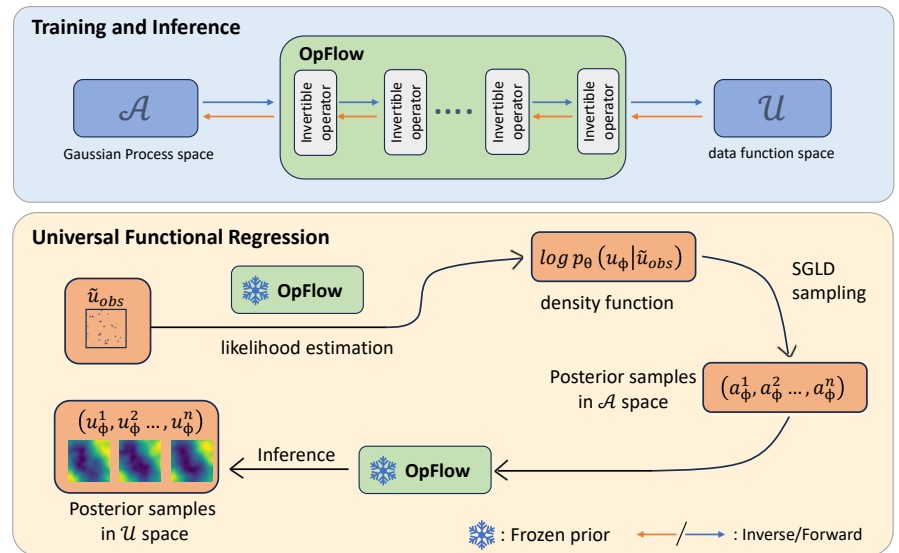

Figure 2: Model architecture of OPFLOW, OPFLOW is composed of a list of invertible operators. For the universal function regression task, OPFLOW is the learnt prior, which is able to provide exact likehood estimation for function point evaluation. $\tilde{u}_{obs}$ is the noisy observation, $u_\phi$ is the posterior function of interest and $u_\phi = \mathcal{G}_\theta(a_\phi)$, where $\mathcal{G}_\theta$ is the learnt forward operator.

Finally, we output the function $(v')^{i+1}$ for the next layer where $(v')^{i+1} = \mathbf{concat}(h_1^i, (h')_2^i)$. The **concat** (concatenation) operation corresponds to the reversing the split operation, and is either in the physical domain or codomain.

**Inverse process.** The inverse process for the $i$the layer of OPFLOW is summarized in Algorithm 3. First, we split the input function $(v')^{i+1}$ with $h_1^i, (h')_2^i = \mathbf{split}((v')^{i+1})$. Then the inverse process of the affine coupling layer also takes $h_1^i$ as input, and generates the scale and shift functions $\log(s^i), b^i = \mathcal{T}_\theta^i(h_1^i)$. Thus, both forward and inverse processes utilize the $h_1^i$ as input for the affine coupling layer, which implies that the scale and shift functions derived from the forward process and inverse process are identical from the affine coupling layer. Then we can reconstruct $h_2^i$ and $v^i$ through $h_2^i = ((h')_2^i - b^i)/s^i$; $v^i = \mathbf{concat}(h_1^i, h_2^i)$. Finally, we have the inverse of the actnorm layer with $(v')^i = (v^i - b_\theta^i)/s_\theta^i$. Similar inverse processes of normalizing flows are described in (Kingma & Dhariwal, 2018; Dinh et al., 2017).

## 4.2 Training

Since OPFLOW is a bijective operator, we only need to learn the inverse mapping $\mathcal{F}_\theta : \mathcal{U} \to \mathcal{A}$, as the forward operator is immediately available. We train OPFLOW in a similar way to other infinite-dimensional generative models (Rahman et al., 2022a; Seidman et al., 2023; Lim et al., 2023) with special addition of domain alignment. The space of functions for $\mathcal{A}$ is taken to be drawn from a GP, which allows for exact likelihood estimation and efficient training of OPFLOW by minimizing the negative log-likelihood. Since OPFLOW is discretization agnostic (Rahman et al., 2022a; Kovachki et al., 2023), it can be trained on various discretizations of $\mathcal{D}_\mathcal{U}$ and be later applied to a new set of discretizations, and in the limit, to the whole of $\mathcal{D}_\mathcal{U}$ (Kovachki et al., 2023). We now define the training objective $\mathcal{L}$ for model training as follows,

$$\mathcal{L} = -\min_{\theta \in \Theta} \mathbb{E}_{u \sim \mathbb{P}_\mathcal{U}}[\log p_\theta(u|_D)], \quad where, \quad \log p_\theta(u|_D) = \log p(a|_{D_\mathcal{A}}) + \sum_{i=1}^{s} \log |\det(\frac{\partial(v^i|_{D_\mathcal{V}^i})}{\partial(v^{i-1}|_{D_\mathcal{V}^{i-1}})})|. \quad (1)$$

The inverse operator is composed of $s$ invertible operator layers with $\mathcal{F}_\theta := \mathcal{F}_\theta^s \circ \mathcal{F}_\theta^{s-1} \cdots \mathcal{F}_\theta^0$, and gradually transforms $v^0$ to $v^1, \cdots v^{s-1}, v^s$, where $v^0 = u, v^s = a$ and the alignment naturally holds within the domains

associated with $v^0, v^1, \cdots v^s$ as each layer of $\mathcal{F}_\theta$ is an invertible operator. $\mathcal{V}^i$ is the collection of positions for function $v^i$ with $D_\mathcal{V}^0 = D$, $D_\mathcal{V}^s = D_\mathcal{A}$, and each data point has its own discretization $D$ [2].

Although the framework of OpFlow is well-defined, the negative log-likelihood for high-dimensional data may suffer from instabilities during training, leading to a failure to recover the true distribution. Through our experiments, we found that relying solely on the likelihood objective in Eq 1 was insufficient for training OpFlow successfully. To address this, we introduce a regularization term, to ensure that OpFlow learns the true probability measure by helping to stabilize the training process and leading to faster convergence. Such regularization is inspired by the infinite-dimensional Wasserstein loss used in (Rahman et al., 2022a). Unlike the implicit Wasserstein loss provided by the discriminator in GANO, we could potentially have a closed-form expression of the infinite-dimensional Wasserstein loss in the context of OpFlow. This is due to the learning process mapping function data onto a well-understood GP. The inherent properties of Gaussian Processes facilitate measuring the distance between the learned probability measure and the true probability measure, which is explained into detail in the subsequent sections.

A common way for measuring the distance between two probability measures is square 2-Wasserstein (Mallasto & Feragen, 2017) distance, which is defined as

$$W_2^2(\mathbb{P}_\mathcal{A}, \mathcal{F}_\theta \sharp \mathbb{P}_\mathcal{U}) = \inf_{\pi \in \Pi} \int_{\mathcal{A} \times \mathcal{A}} d_2^2(a_1, a_2) d\pi, \ (a_1, a_2) \in \mathcal{A} \times \mathcal{A}, \tag{2}$$

where $\Pi$ is a space of joint measures defined on $\mathcal{A} \times \mathcal{A}$ such that for any $\pi \in \Pi$, the margin measures of $\pi(a_1, a_2)$ on the first and second arguments are $\mathbb{P}_\mathcal{A}$ and $\mathcal{F}_\theta \sharp \mathbb{P}_\mathcal{U}$, and $d_2$ is a metric induced by the $L^2$ norm. By construction, $\mathcal{A} \sim \mathcal{GP}$ and if sufficiently trained, $\mathcal{F}_\theta \sharp \mathcal{U} \sim \mathcal{GP}$. We can thus further simplify Eq. 2, which is named as $F^2ID$ score by Rahman et al. (2022b), and is defined as:

$$W_2^2(\mathbb{P}_\mathcal{A}, \mathcal{F}_\theta \sharp \mathbb{P}_\mathcal{U}) = d_2^2(m_1, m_2) + Tr(\mathcal{K}_1 + \mathcal{K}_2 - 2(\mathcal{K}_1^{\frac{1}{2}} \mathcal{K}_2 \mathcal{K}_1^{\frac{1}{2}})^{\frac{1}{2}}). \tag{3}$$

for which $\mathbb{P}_\mathcal{A}$ is chosen to be $\mathcal{GP}_1(m_1, \mathcal{K}_1)$ and $\mathcal{GP}_2(m_2, \mathcal{K}_2)$ is the GP fit to the push-forward measure $\mathcal{F}_\theta \sharp \mathbb{P}_\mathcal{U}$. The $m_1, \mathcal{K}_1$ are the mean function and covariance operator of $\mathcal{GP}_1(m_1, \mathcal{K}_1)$, whereas $m_2, \mathcal{K}_2$ are the mean function and covariance operator of $\mathcal{GP}_2(m_2, \mathcal{K}_2)$. We note that it is equivalent to say a function $f \sim \mathcal{GP}(m, k)$ or $f \sim \mathcal{GP}(m, \mathcal{K})$, where $k$ is the covariance function and $\mathcal{K}$ is the covariance operator (Mallasto & Feragen, 2017). We take Eq. 3 as the regularization term. Furthermore, we inject noise with negligible amplitude that sampled from $\mathcal{GP}_1$ to the training dataset $\{u_j\}_{j=1}^m$. In this way, we can have the inversion map to be well-define and stable. From here, we can define a second objective function.

$$\mathcal{L} = -\min_{\theta \in \Theta} \mathbb{E}_{u \sim \mathbb{P}_u}[\log p_\theta(u|D)] + \lambda W_2^2(\mathbb{P}_\mathcal{A}|_D, \mathcal{F}_\theta \sharp \mathbb{P}_\mathcal{U}|_D), \tag{4}$$

where $\lambda$ is the weight for the regularization term. For the collections of position $D_\mathcal{A} = \{y_i\}_{i=1}^l, y_i \in \mathcal{D}_\mathcal{A}$, we select a finite set of orthonormal basis $\{e_1, e_2, ..e_l\}$ on the $L^2(\mathcal{D}_\mathcal{A})$ space such that $\|e_i\|_2 = \frac{1}{\sqrt{l}}, i \in \{1, ..l\}$. Thus the relationship between the covariance operator and the covariance function form can be expressed as:

$$< \mathcal{K}e_i, e_j > = \int_{\mathcal{D}_\mathcal{A}} \frac{1}{\sqrt{l}} k(y, y_i) e_j(y) dy = \frac{1}{l} k(y_i, y_j) \tag{5}$$

Thus, given the chosen set of orthonormal basis functions, the covariance matrix (covariance function in matrix form) $K$ associated with the covariance operator $\mathcal{K}$ follows the relationship:

$$K[i, j] = l < \mathcal{K}e_i, e_j > = k(y_i, y_j) \tag{6}$$

---

[2]Unlike finite-dimensional normalizing flows on fixed-size and regular grids, OpFlow can provide likelihood estimation $\log p_\theta(u|D)$ on arbitrary grids $D$, which represents a potentially irregular discretization of $\mathcal{D}_\mathcal{U}$. To compute $\log p_\theta(u|D)$, we need the homeomorphic mapping between $\mathcal{D}_\mathcal{A}$ and $\mathcal{D}_\mathcal{U}$ to find $D_\mathcal{A}$, corresponding to $D$. The density function in Eq. 1 holds for new points of evaluation. Our implementation of OpFlow incorporates FNO that requires inputs on regular grids. Consequently, although OpFlow is designed to handle functions for irregular grids, our specific implementation of OpFlow only allows for likelihood estimation on regular girds with arbitrary resolution.

Where, $K[i,j]$ is the $(i_{th}, j_{th})$ component of the covariance matrix $K$. Detailed explanation of the chosen basis function can be found in the Appendix A.1 in (Rahman et al., 2022b). Thus the matrix form of the Eq. 3 can be expressed as follows,

$$\frac{1}{l}\|h_1 - h_2\|_2^2 + \frac{1}{l}Tr(K_1 + K_2 - 2(K_1^{\frac{1}{2}}K_2K_1^{\frac{1}{2}})^{\frac{1}{2}}) = \frac{1}{l}(\|h_1 - h_2\|_2^2 + \|K_1^{\frac{1}{2}} - K_2^{\frac{1}{2}}\|_f^2) \tag{7}$$

and $h1, h2$ are the matrix forms of means $m_1, m_2$ in Eq. 3, $K_1, K_2$ are the covariance matrix, corresponding to $\mathcal{K}_1$ and $\mathcal{K}_2$. $\|\|_2^2$ is the square of $L^2$ norm and $\|\|_f^2$ is the square of Frobenius norm. For detailed proof of the simplification of the trace of covariance matrix for two Gaussian processes in Eq. 7, please refer to (Givens & Shortt, 1984). Equipped with the above preliminary knowledge, we introduce a two-step strategy for training as shown in Algorithm 1. In the first step, we use the objective defined in Eq. 4 with $\lambda > 0$, whereas in the second step, we fine-tune the model by setting $\lambda = 0$ and a smaller learning rate is preferred. For step one, even though we have the closed and simplified form of the matrix form of square 2-Wasserstein distance shown in Eq. 7, calculating the differentiable square root of the matrix is still time consuming and may be numerically unstable (Pleiss et al., 2020). Here, we directly calculate $\|K_1 - K_2\|_f^2$ instead of $\|K_1^{\frac{1}{2}} - K_2^{\frac{1}{2}}\|_f^2$, utilizing $\widehat{W}_2^2 = \frac{1}{l}(\|h_1 - h_2\|_2^2 + \|K_1 - K_2\|_f^2)$, and such approximation is acceptable due to the the regularization term only exists in the warmup phase and is not a tight preparation. In the appendix, we show an ablation study that demonstrates the need for the regularization in the warmup phase. Furthermore, Eq. 7 implies we normalize the Wasserstein distance and makes sure the square of $L^2$ norm and Frobenius norm remain finite as the number of discretization points increases.

---

**Algorithm 1** OPFLOW Training

---

1: **Input:** Training dataset $\{u_j|_D\}_{j=1}^m$; collection of positions $D = \{x_i, \cdots x_l\}$; mean vector $h_1$ and covariance matrix $K_1$; noise level $\gamma$; random noise samples $\{\nu_i\}_{i=1}^n \sim \mathcal{GP}_1$; regularization weight $\lambda$

2: **Initialize the reverse OpFlow** $\mathcal{F}_\theta$

3: **while** warmup **do**

4:     Random batch $\{u_i|_D\}_{i=1}^n$ from $\{u_j|_D\}_{j=1}^m$

5:     Inject $\mathcal{GP}$ noise: $\{u_i|_D\}_{i=1}^n = \{u_i|_D + \gamma \cdot \nu_i|_D\}_{i=1}^n$

6:     Construct $h_2, K_2$: $h_2 = \frac{1}{n}\sum_{i=1}^n \mathcal{F}_\theta(u_i|_D)$, $K_2 = cov(\{\mathcal{F}_\theta(u_i|_D)\}_{i=1}^n)$

7:     Calculate $W_2^2$ approximate regularization term: $\widehat{W}_2^2 = \frac{1}{l}(\|h_1 - h_2\|_2^2 + \|K_1 - K_2\|_f^2)$

8:     Compute the raining objective $\mathcal{L} = -\frac{1}{n}\sum_i^n \log p_\theta(u_i|_D) + \lambda\widehat{W}_2^2$

9:     Update $\mathcal{F}_\theta$ through gradient descent of $\mathcal{L}$

10: **end while**

11: **while** finetune **do**

12:     Random batch $\{u_i|_D\}_{i=1}^n$ from $\{u_j|_D\}_{j=1}^m$

13:     Inject $\mathcal{GP}$ noise: $\{u_i|_D\}_{i=1}^n = \{u_i|_D + \gamma \cdot \nu_i|_D\}_{i=1}^n$

14:     Computing training objective $\mathcal{L} = -\frac{1}{n}\sum_i^n \log p_\theta(u_i|_D)$

15:     Update $\mathcal{F}_\theta$ through gradient descent of $\mathcal{L}$

16: **end while**

---

### 4.3 Universal Functional Regression with OpFlow

Bayesian functional regression treats the mapping function itself as a stochastic process, placing a prior distribution over the mapping function space. This allows for predictions with uncertainty estimates throughout $\mathcal{D}$, most importantly away from the observation points. GPR places a GP prior on the mapping function space, a common choice because of mathematical tractability. However, there are many real world datasets and scenarios that are not well-represented by a GP (Kındap & Godsill, 2022). To address this, we outline an algorithm for UFR with OPFLOW in Bayesian inference problems.

In the following we assume that OPFLOW has been trained previously and sufficiently approximates the true stochastic process of the data. We thus have an invertible mapping between function spaces $\mathcal{U}$ and $\mathcal{A}$. We now can use the trained OPFLOW as a learned prior distribution over the space of possible mapping functions, fixing the parameters of OPFLOW for functional regression.

Similar to GPR, suppose we are given $\tilde{u}_{obs}$, pointwise and potentially noisy observations of a function $u$, where the observation points are at the discretization set $D \subset \mathcal{D}_{\mathcal{U}}$. We refer to the evaluations at $D$ as $u_{obs}$, and the additive noise $\epsilon$ is the white noise with the law $\epsilon \sim \mathcal{N}(0, \sigma^2)$. Note that this is based on the conventions in GPRs; otherwise, the noise $\epsilon$ can also be treated as a function with a GP from. One more time, similar to GPR, we aim to infer the function's values on a set of new points $D'$ where $D \subset D'$ given the observation $\tilde{u}_{obs}$. The regression domain therefore is $D'$ at which we refer to the inferred function's values as $u_\phi$ and $a_\phi = \mathcal{F}_\theta(u_\phi)$. Ergo, for the likelihood of the values on points in $D'$, i.e., the likelihood of $u_\phi$, we have,

$$\log p_\theta(u_\phi | \tilde{u}_{obs}) = \log p_\theta(\tilde{u}_{obs} | u_\phi) + \log p_\theta(u_\phi) - \log p_\theta(\tilde{u}_{obs}) \tag{8}$$

$$= \log p_\theta(\tilde{u}_{obs} | u_{\phi|D}) + \log p_\theta(u_\phi) - \log p_\theta(\tilde{u}_{obs}) \tag{9}$$

$$= -\frac{\|\tilde{u}_{obs} - u_{\phi|D}\|_2^2}{2\sigma^2} + \log p_\theta(u_\phi) - r \log \sigma - \frac{1}{2} r \log(2\pi) - \log p_\theta(\tilde{u}_{obs}), \tag{10}$$

where $p_\theta(u|_D)$ denotes the probability of $u$ evaluated on the collection of points $D$ with the learned prior (OPFLOW), and $r$ is the cardinality of set $D$ with $r = |D|$. Maximizing the above equation results in maximum *a posteriori* (MAP) estimate of $p_\theta(u_\phi | \tilde{u}_{obs})$. Since the last three terms in Eq.10 are constants, such maximization is carried using gradient descent focusing only on the first two terms. In the following, we refer to $\overline{u}_\phi$ as the MAP estimate given $\tilde{u}_{obs}$.

For sampling for the posterior in Eq. 10, we utilize the fact that OPFLOW is a bijective operator and the target function $u_\phi$ uniquely defines $a_\phi$ in $\mathcal{A}$ space, therefore, drawing posterior samples of $u_\phi$ is equivalent to drawing posterior samples of $a_\phi$ in Gaussian space where SGLD with Gaussian random field perturbation is relevant. Our algorithm for sampling from the $p_\theta(u_\phi | \tilde{u}_{obs})$ is described in Algorithm 2. We initialize $a_\phi | \tilde{u}_{obs}$ with the MAP estimate of $p_\theta(u_\phi | \tilde{u}_{obs})$, i.e., $\overline{u}_\phi$, and follow Langevin dynamics in $\mathcal{A}$ space with Gaussian structure to sample $u_\phi$. This choice helps to stabilize the functional regression and further reduces the number of burn-in iterations needed in SGLD. The burn-in time $b$ is chosen to warm up the induced Markov process in SGLD and is hyperparameter to our algorithm (Ahn, 2015). We initialize the SGLD with $a_\phi^0 = \mathcal{F}_\theta(\overline{u}_\phi)$ and $u_\phi^0 = \mathcal{G}_\theta(a_\phi^0) = \overline{u}_\phi$. Then, at each iteration, we draw a latent space posterior sample $a_\phi^t$ in $\mathcal{A}$ space, where $a_\phi^t$ is the $t^{th}$ sample of function $a_\phi$, and harvest the true posterior samples $u_\phi^t$ by calling $\mathcal{G}_\theta(a_\phi^t)$. Since $p_\theta(a_\phi | \tilde{u}_{obs}) \sim \mathcal{GP}$, it is very efficient to sample from it, and we find that relatively little hyperparameter optimization is needed. Since the sequential samples are correlated, we only store samples every $t_N$ time step, resulting in representative samples of the posterior without the need to store all the steps of SGLD (Riabiz et al., 2022).

---

**Algorithm 2** Sample from posterior using SGLD
---

1: **Input and Parameters:** OPFLOW models $\mathcal{G}_\theta, \mathcal{F}_\theta$, Langevin dynamics temperature $T$, $\mathcal{G}_\theta$, MAP $\overline{u}_\phi$.
2: **Initialization**: $a_\phi^0 = \mathcal{F}_\theta(\overline{u}_\phi)$ and $u_\phi^0 = \mathcal{G}(a_\phi^0)$,
3: **for** $t = 0, 1, 2, \ldots, N$ **do**
4:      Compute gradient of the posterior: $\nabla_{a_\phi} \log p_\theta(u_\phi^t | \tilde{u}_{obs})$
5:      Update $a_\phi^{t+1}$: $a_\phi^{t+1} = a_\phi^t + \frac{\eta_t}{2} \nabla \log p_\theta(u_\phi^t | \tilde{u}_{obs}) + \sqrt{\eta_t T} \mathcal{N}(0, I)$
6:      **if** $t \geq b$ **then**
7:          Every $t_N$ iterations: obtain new sample $a_\phi^{t+1}$, $u_\phi^{t+1} = \mathcal{G}_\theta(a_\phi^{t+1})$
8:      **end if**
9: **end for**

---

# 5 Experiments

In this section, we aim to provide a comprehensive evaluation of OPFLOW's capabilities for UFR, as well as generation tasks. We consider several ground truth datasets, composed of both Gaussian and non-Gaussian processes, as well as a real-world dataset of earthquake recordings with highly non-Gaussian characteristics. [3]

---

[3] https://github.com/yzshi5/OpFlow

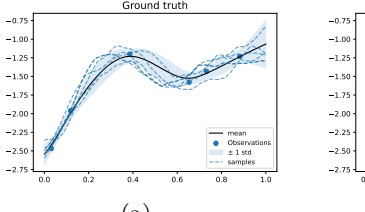 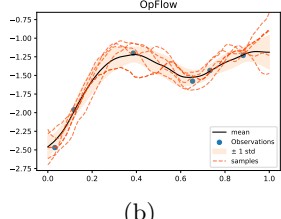 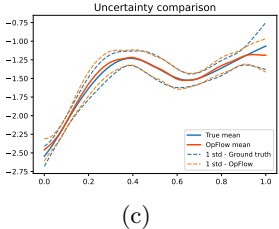

(a)  (b)  (c)

Figure 3: OpFlow regression on GP data. (a) Ground truth GP regression with observed data and predicted samples (b) OpFlow regression with observed data and predicted samples. (c) Uncertainty comparison between true GP and OpFlow predictions.

## 5.1 Universal Functional Regression

**Summary of parameters.** Here, we show UFR experiments using the domain decomposed implementation of OpFlow, and leave the codomain OpFlow UFR experiments for the appendix. Table 2 lists the UFR hyperparameters used, as outlined in Algorithm 2. For training OpFlow, we take $\mathcal{A} \sim \mathcal{GP}()$ with a Matern covariance operator parameterized by length scale $l$ and roughness $\nu$. The mean function for the $\mathcal{GP}$ is set to zero over $\mathcal{D}$. The noise level ($\sigma^2$) of observations in Eq. 10 is assumed as 0.01 for all regression tasks.

**Gaussian Processes.** This experiment aims to duplicate the results of classical GPR, where the posterior is known in closed form, but with OpFlow instead. We build a training dataset for $\mathcal{U} \sim \mathcal{GP}$, taking $l_{\mathcal{U}} = 0.5, \nu_{\mathcal{U}} = 1.5$ and the training dataset contains 30000 realizations. For the latent space GP, we take $l_{\mathcal{A}} = 0.1, \nu_{\mathcal{A}} = 0.5$. The resolution of $\mathcal{D}$ is set to 128 for the 1D regression experiment. Note that, the OpFlow is not explicitly notified with the mean and covariance of the data GP, and learns them using samples, while the GP ground truth is based on known parameters.

For performing the regression, we generate a single new realization of $\mathcal{U}$, and select observations at 6 random positions in $\mathcal{D}$. We then infer the posterior for $u$ across the 128 positions, including noise-free estimations for the observed points. Fig. 3 displays the observation points and analytical solution for GP regression for this example, along with the posterior from OpFlow. In general, the OpFlow solution matches the analytical solution well; the mean predictions of OpFlow (equivalent to the MAP estimate for GP data) and its assessed uncertainties align closely with the ground truth. This experiment demonstrates OpFlow's ability to accurately capture the posterior distribution, for a scenario where the observations are from a Gaussian Process with an exact solution available.

Furthermore, we present the regression results of OpFlow compared with other Deep GP models in Appendix A.4. In this comparison, OpFlow exhibits superior performance over conventional Deep GP approach. Last, we explore the minimal size of the training dataset to train an effective prior. In Appendix A.11, we detail our analysis of the 1D GP regression experiment using OpFlow with reduced training dataset sizes. From this investigation, we conclude that the training dataset containing between 3000 and 6000 samples (10% to 20% of the original dataset size) is sufficient to train a robust and effective prior.

**Truncated Gaussian Processes.** Next, we apply OpFlow to a non-Gaussian functional regression problem. Specifically, we use the truncated Gaussian process (TGP) (Swiler et al., 2020) to construct a 1D training dataset. The TGP is analogous to the truncated Gaussian distribution, which bounds the functions to have amplitudes lying in a specific range. For the dataset, we take $l_{\mathcal{U}} = 0.5$ and $\nu_{\mathcal{U}} = 1.5$, and truncate the function amplitudes to the range $[-1.2, 1.2]$. For the latent space GP, we take $l_{\mathcal{A}} = 0.1, \nu_{\mathcal{A}} = 0.5$. The resolution is set to 128. We conduct an experiment with only 4 random observation points, infer the function across all 128 positions. Unlike most non-Gaussian processes, where the true posterior is often intractable, the TGP framework, governed by parameters $l_{\mathcal{U}}, \nu_{\mathcal{U}}$ and known bounds, allows for true posterior estimation using rejection sampling. We consider a narrow tolerance value of $\pm 0.10$ around each observation. The tolerance range can be analogous to but conceptually distinct from noise level (Robert et al., 1999; Swiler et al., 2020). For the training dataset, we use 30000 realizations of the TGP. For the posterior evaluation, we generate 1000 TGP posterior samples, and taken the calculated mean and uncertainty from the posterior samples as the ground truths.

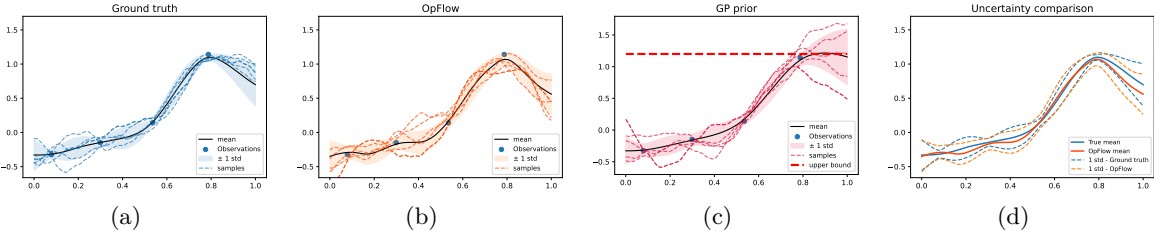

Figure 4: OPFLOW regression on TGP data. (a) Ground truth TGP regression with observed data and predicted samples (b) OPFLOW regression with observed data and predicted samples. (c) prior GP regression with observed data and predicted samples (c) Uncertainty comparison between true TGP and OPFLOW predictions.

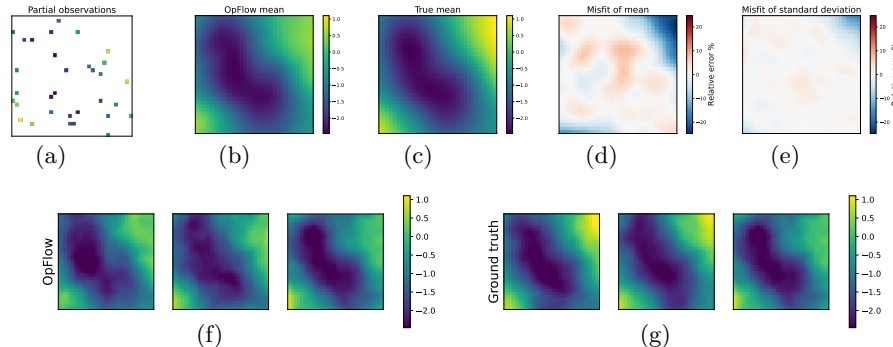

Figure 5: OPFLOW regression on 32×32 GRF data. (a) 32 random observations. (b) Predicted mean from OPFLOW. (c) Ground truth mean from GP regression. (d) Misfit of the predicted mean. (e) Misfit of predicted uncertainty. (f) Predicted samples from OPFLOW. (g) Predicted samples from GP regression.

Fig. 4 shows the UFR results for this dataset alongside the true TGP posterior. We see that OPFLOW effectively captures the true non-Gaussian posterior, producing samples that are visually aligned with the ground truth. For comparison, we also show the results of applying GP regression to this dataset. Notice that the GP prior allows for non-zero probability outside of the truncation range, whereas the OPFLOW prior learns these constraints, resulting in a much more accurate posterior. Fig. 4 reveals a slight deviation in OPFLOW's mean prediction starting from $x = 0.8$, yet the uncertainty remains accurately represented. This discrepancy may be explained by the challenging nature of this task, which involves using only 4 observations. With more observations, OPFLOW's predictions are expected to improve.

**Gaussian random fields.** We build a 2D dataset of Gaussian random fields having $l_{\mathcal{U}} = 0.5$ and $\nu_{\mathcal{U}} = 1.5$. For the latent space, we have a GP with $l_{\mathcal{A}} = 0.1, \nu_{\mathcal{A}} = 0.5$. We set the resolution of the physical domain to $32 \times 32$ for the training data. We generate 20000 samples in total for training. This is another experiment with an analytical expression available for the true posterior. We generate one new realization of the data for functional regression, and choose observations at 32 random positions, as shown in Fig 5a. We then infer the possible outcomes over the entire $32 \times 32$ grid. Fig 5 shows that OPFLOW reliably recovers the posterior without assuming beforehand that the true data are from a Gaussian process. The error is non-uniform over the domain, the largest of which occurs in the data-sparse upper right corner. Fig 5 shows several samples of OPFLOW's posterior, mirroring the true posterior variability produced by GPR. We show a second scenario for this same function in the appendix, but where the observations are concentrated along linear patterns (Fig 11).Similarly, OPFLOW faithfully approximates the true posterior.

**Seismic waveforms regression.** In this experiment, OPFLOW regression is applied to a real-world non-Gaussian dataset consisting of earthquake ground motion time series records from Japan Kiban–Kyoshin network (KiK-net). The raw data are provided by the Japanese agency, National Research Institute for Earth Science and Disaster Prevention (NIED, 2013). The KiK-net data encompasses 20,643 time series of ground velocity for earthquakes with magnitudes in the range 4.5 to 5, recorded between 1997 and 2017. All seismic

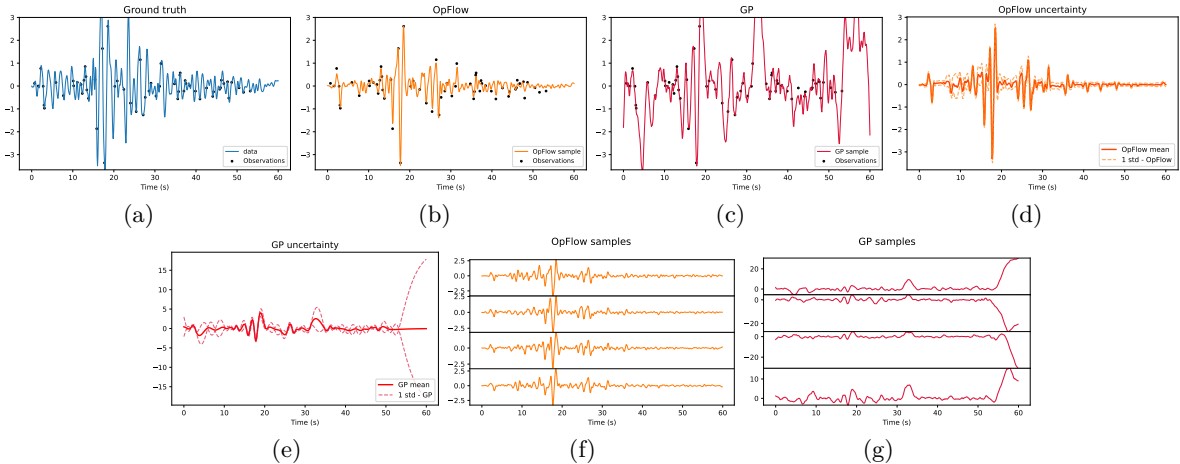

Figure 6: OPFLOW regression on seismic waveform data. (a) Ground truth waveform. (b) One predicted sample from OPFLOW. (c) One predicted sample from the best fitted GP. (d) Predicted mean and uncertainty from OPFLOW. (e) Predicted mean and uncertainty from the best fitted GP. (f) Samples from OPFLOW. (g) Samples from the best fitted GP.

signals were resampled at 10Hz and have a 60 second duration, resulting in a resolution of 600 for our analysis. The time series are aligned such that $t = 0$ coincides with the onset of shaking. A second-order Gaussian filter was used to attenuate high-frequency wiggles. We used $l_{\mathcal{A}} = 0.05, \nu_{\mathcal{U}} = 0.5$ for the latent Gaussian space $\mathcal{A}$. After training the OPFLOW prior, we select a new (unseen) time series randomly for testing (Fig. 6a), and perform UFR over the entire domain with 60 randomly selected observation points, (Fig. 6). For comparison, we show the results for GP regression with parameters optimized via likelihood maximization. Fig 6b shows that a sample from OPFLOW matches the actual data, with OPFLOW accurately capturing key characteristics in the data that include various types of seismic waves arriving at different times in the time series (Shearer, 2019). Conversely, the standard GP model struggles with the complexity of the seismic data as shown in Fig. 6c. Further, Fig. 6d,e illustrate that OPFLOW provides a reasonable estimation of the mean and uncertainty, whereas the GP model's uncertainty significantly increases after 50 seconds due to a lack of observations. Additional samples from OPFLOW and GP (Fig. 6f,g) further highlight the superior performance of OPFLOW for functional regression of seismic waveforms.

## 5.2 Generation Tasks

While our emphasis in this study is on functional regression, there is still value in evaluating the generative capabilities of OPFLOW. Here, we take GANO as a baseline model as it is a generative neural operator with discretization invariance. The following study demonstrates that domain decomposed OPFLOW provides comparable generation capabilities similar to prior works, such as GANO. The implementation of GANO used herein follows that of the paper (Rahman et al., 2022a). In the following experiments, the implementations of the GP for the Gaussian space $\mathcal{A}$ of GANO is adapted to optimize GANO's performance for each specific case. Unless otherwise specified, the default GP implementation utilize the Gaussian process package available in Scikit-learn (Pedregosa et al., 2011). More generation experiments of domain decomposed OPFLOW and codomain OPFLOW are provided in the appendix.

**GP data.** For both OPFLOW and GANO, we set Matern kernel parameters with $l_{\mathcal{U}} = 0.5, \nu_{\mathcal{U}} = 1.5$ for the observed data space. And $l_{\mathcal{A}} = 0.05, \nu_{\mathcal{A}} = 0.5$ for the latent Gaussian space of OPFLOW, $\alpha = 1.5, \tau = 1.0$ for the latent Gaussian space of GANO, where $\alpha, \tau$ are the coefficient parameter and inverse length scale parameter of the specific implementation of Matérn based Gaussian process used in GANO paper (Rahman et al., 2022a; Nelsen & Stuart, 2021). Data resolution used here is 256. Fig. 7 shows that OPFLOW can generate realistic samples, virtually consistent with the true data. In contrast, GANO's samples exhibit small high frequency artifacts. Both OPFLOW and GANO successfully recover the statistics with respect to the autovariance function and histogram of the point-wise value (amplitude distribution). We note that there is

a significant difference in model complexity here, as OPFLOW has 15M parameters whereas GANO has only 1.4M. The appendix shows a super-resolution experiment for this dataset.

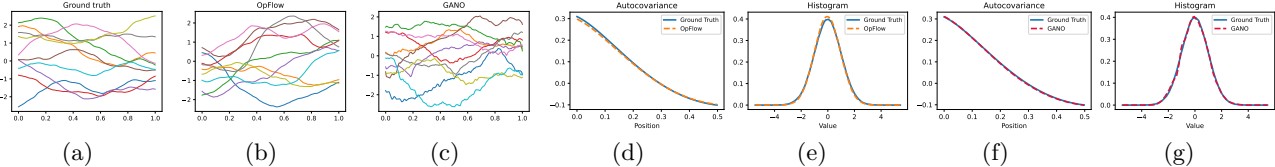

Figure 7: OPFLOW generation performance on GP data. (a) Samples from the ground truth. (b) Samples from OPFLOW. (c) Samples from GANO. (d) Autocovariance of samples from OPFLOW. (e) Histogram of samples from OPFLOW. (f) Autocovariance of samples from GANO. (g) Histogram of samples from GANO.

**TGP data.** We maintain the same parameter settings as used in the TGP functional regression experiment, with bounds set between -1.2 and 1.2, and Matern kernel parameters $l_{\mathcal{U}} = 0.5, \nu_{\mathcal{U}} = 1.5$ for the observed space and $l_{\mathcal{A}} = 0.1, \nu_{\mathcal{A}} = 0.5$ for the latent space of OPFLOW, $\alpha = 1.5, \tau = 1.0$ for the latent Gaussian space of GANO. The only difference is that we increase the resolution to 256 from 128. Fig. 8 demonstrates that OPFLOW is capable of generating realistic samples that closely mimic the ground truth samples, while GANO tends to produce samples that appear rougher in texture. Details on the super-resolution experiment for this task are available in the appendix.

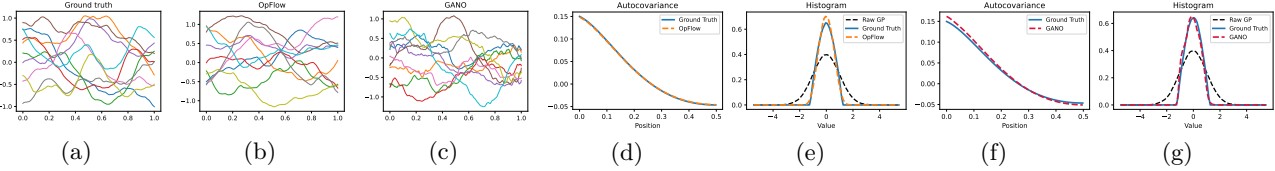

Figure 8: OPFLOW generation performance on TGP data. (a) Samples from the ground truth. (b) Samples from OPFLOW. (c) Samples from GANO. (d) Autocovariance of samples from OPFLOW. (e) Histogram of samples from OPFLOW. (f) Autocovariance of samples from GANO. (g) Histogram of samples from GANO.

**Seismic Data.** The dataset is the same as used in the functional regression section. However for this test, we no longer use a Gaussian filter to simplify the training dataset, which makes it more challenging to learn. The results are shown in Fig. 9. These figures demonstrate that both OPFLOW and GANO can generate visually realistic seismic waveforms. For validation purposes, we examine the Fourier Amplitude Spectrum, a critical engineering metric (Shi et al., 2024b), in Fig. 9. The analysis shows that both models accurately learn the frequency content of the seismic velocity dataset, underscoring their potential utility in seismic hazard analysis and engineering design (Shi et al., 2024b).

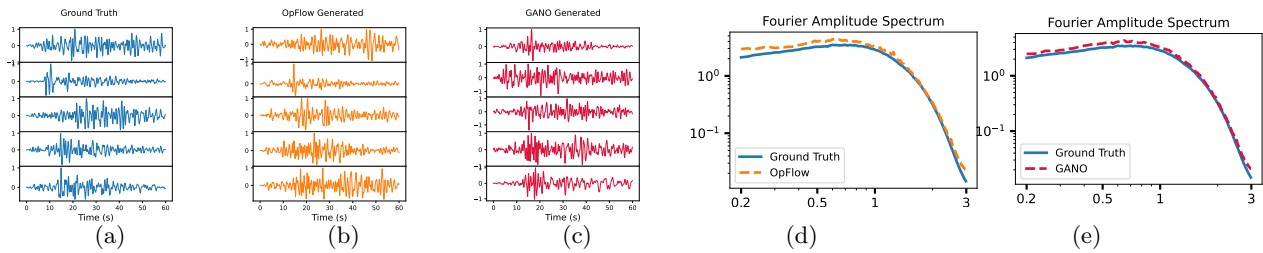

Figure 9: OPFLOW generation performance on seismic velocity data from kik-net. (a) Samples from the ground truth. (b) Samples from OPFLOW. (c) Samples from GANO. (d) Fourier amplitude spectrum of samples from OPFLOW. (e) Fourier amplitude spectrum of samples from GANO

## 6 Conclusion

In this paper, we have introduced Neural Operator Flows (OPFLOW), an invertible infinite-dimensional generative model that allows for universal functional regression. OPFLOW offers a learning-based solution

for end-to-end universal functional regression, which has the potential to outperform traditional functional regression with Gaussian processes. This is in addition to OPFLOW comparing favorably with other algorithms for generation tasks on function spaces. However, we acknowledge that our research primarily focuses on 1D and 2D datasets. Learning operators for functions defined on high-dimensional domains remains challenging and is an area that has not been thoroughly developed. Looking ahead, OPFLOW may allow for new avenues in analysis of functional data by learning directly from the data, rather than assuming the type of stochastic process beforehand.

## Acknowledgements

The authors would like to express their gratitude to the TMLR reviewers and the action editor for their constructive comments. We also extend our thanks to Chenxi Kong for providing the data for the Rayleigh-Bénard convection problem. Z. Ross was supported by a fellowship from the David and Lucile Packard Foundation.

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

# A Appendix

## A.1 Forward and Inverse processes of OpFlow

---
**Algorithm 3** Forward and Inverse processes of OPFLOW
---

1: **Input and Parameters:** Input function $(v')^i$ to the $i$th layer of OPFLOW; $i$th actnorm layer parameterized by $s_\theta^i, b_\theta^i$; $i$th affine coupling layer $\mathcal{T}_\theta^i$; Discretized domain $D_\theta$ for $s_\theta^i, b_\theta^i$ and $D_s$ for the output functions of the affine coupling layer

2: **Forward process:**

3:      Input function: $(v')^i$

4:      Actnorm layer: $v^i = s_\theta^i \odot (v')^i + b_\theta^i$

5:      Affine coupling layer: $h_1^i, h_2^i = \mathbf{split}(v^i)$; $\log(s^i), b^i = \mathcal{T}_\theta^i(h_1^i)$; $(h')_2^i = s^i \odot h_2^i + b^i$

6:      Output function: $(v')^{i+1} = \mathbf{concat}(h_1^i, (h')_2^i)$

7: **Inverse process:**

8:      Input function: $(v')^{i+1}$

9:      Partition: $h_1^i, (h')_2^i = \mathbf{split}((v')^{i+1})$

10:      Inverse of the affine coupling layer: $\log(s^i), b^i = \mathcal{T}_\theta^i(h_1^i)$; $h_2^i = ((h')_2^i - b^i)/s^i$; $v^i = \mathbf{concat}(h_1^i, h_2^i)$

11:      Inverse of the actnorm layer: $(v')^i = (v^i - b_\theta^i)/s_\theta^i$

12:      Output function: $(v')^i$

13: **Log-determinant:**

14:      Actnorm layer: $\mathbf{sum}(\log(s_\theta^i|_{D_\theta}))$

15:      Affine coupling layer: $\mathbf{sum}(\log(s^i|_{D_s}))$

---

## A.2 Discretization agnostic property of OpFlow

In this part, we first expand the definition of neural operators and elaborate on their discretization-invariant properties, then explain why OPFLOW is also discretization invariant.

Given two separable Banach spaces $\mathcal{A}$ and $\mathcal{U}$, a neural operator (denoted as $\mathcal{G}_\theta$) learns mappings between two function spaces. $\mathcal{G}_\theta : \mathcal{A} \to \mathcal{U}$, involving input and output functions on bounded domain in real-valued spaces. Thus for an input function $a \in \mathcal{A}$, we have $\mathcal{G}_\theta(a) = u$, where $u \in \mathcal{U}$. The architecture of neural operators is usually composed of three parts (Kovachki et al., 2023) : (1) **Lifting:** Converts input functions $a : \mathcal{D} \to \mathbb{R}^{d_a}$ into a higher-dimensional space $\mathbb{R}^{d_{v_0}}$ using a local operator, with $d_{v_0} > d_a$. (2) **Kernel Integration:** For $i = 0$ to $T - 1$, map hidden state $v_i$ to the next $v_{i+1}$ by a combination of local linear, non-local integral operator and a bias function. (3) **Projection:** The final hidden state $v_T$ is projected to the output space $\mathbb{R}^{d_u}$ through a local operator, where $d_{v_T} > d_u$.

The lift and projection parts in neural operators are usually shallow neural networks, such as Multilayer Perceptrons. For $i$th kernel integration layer, we have the integral kernel operator $\mathcal{K}_i$ with corresponding kernel function $k_i \in C(\mathcal{D}_{i+1} \times \mathcal{D}_i; \mathbb{R}^{d_{v_{i+1}} \times d_{v_i}})$, where $v_{i+1}(y), v_i(x)$ are defined on domains $\mathcal{D}_{i+1}, \mathcal{D}_i$, respectively. Denote $\mu_i$ as the measure on $\mathcal{D}_i$, $W_i$ as the local linear opeartor, $b_i$ as the bias function. Then for the $i$th kernel integration layer, we have

$$v_{i+1}(y) = \int_{D_i} \kappa_i(x, y)v_i(x)d\mu_i(x) + W_i v_i(y) + b_i(y), \quad \forall y \in \mathcal{D}_{i+1} \tag{11}$$

Fourier Neural Operator (Li et al., 2021) simplifies Eq 11 using Fourier transform and achieves quasi-linear time complexity, which can be expressed as

$$v_{i+1} = \mathcal{F}^{-1}(R_i(\mathcal{F}v_i)) + W_i v_i + b_i \tag{12}$$

where $R_i$ are learnable parameters defined in Fourier space associated with user-defined Fourier modes, $\mathcal{F}$ and $\mathcal{F}^{-1}$ are Fourier transformation and inverse Fourier transformation respectively. The calculation of Eq 11,12 for data with arbitrary discretization involves Riemann summation. For more details of calculation of the kernel integration, please refer to Kovachki et al. (2023); Li et al. (2021); Rahman et al. (2022a).

Furthermore, the formulation of OPFLOW is for function spaces, all components and operations invovled in OPFLOW are directly defined on function space. The likelihood computation in Eq. 1 can take place

for any discretization, and moreover, the 2-Wasserstein distance in Eq. 2 is derived for function spaces, which we show how to compute on given discretizations in Eq. 7. Therefore OPFLOW is discretization agnostic. We further use a series of zero-shot super-resolution experiments in Appendix A.7 to demonstrate the discretization-invariant feature of OPFLOW.

### A.3 Parameters used in regression tasks

Table 2 shows the regression parameters (explained in Algorithm 2) for all regression experiments used in this paper. The learning rate is decayed exponentially, which starts from $\{\eta_t, t = 0\}$, ends with $\{\eta_t, t = N\}$

Table 2: Datasets and regression parameters

| Datasets | total iterations ($N$) | burn-in iterations ($b$) | sample iterations ($t_N$) | temperature of noise ($T$) | initial learning rate ($\eta_t, t = 0$) | end learning rate ($\eta_t, t = N$) |
|---|---|---|---|---|---|---|
| GP | 4e4 | 2e3 | 10 | 1 | 5e-3 | 4e-3 |
| TGP | 4e4 | 2e3 | 10 | 1 | 5e-3 | 4e-3 |
| GRF | 2e4 | 2e3 | 10 | 1 | 5e-3 | 4e-3 |
| Seismic | 4e4 | 2e3 | 10 | 1 | 1e-3 | 8e-4 |
| Codomain GP | 4e4 | 2e3 | 10 | 1 | 5e-5 | 4e-5 |

### A.4 Comparison against Deep GPs on Gaussian Process example

In this experiment, we evaluate the regression performance of OPFLOW against that of variational Deep GP (Salimbeni & Deisenroth, 2017) and Deep Sigma Point Processes (DSSP) (Jankowiak et al., 2020). We maintain the same dataset and hyperparameter settings for OPFLOW as those used in the Gaussian process regression example discussed in the main text. The only difference in this experiment is that it includes 10 observations. For the baseline models, we use official implementations from GPyTorch (Gardner et al., 2021) and present the best available results.

For this comparison, we employ the following metrics: (1) Standardized Mean Squared Error (SMSE), which normalizes the mean squared error by the variance of the ground truth; and (2) Mean Standardized Log Loss (MSLL), first proposed in Equation 2.34 of Rasmussen & Williams (2006), defined as:

$$-\log p(\hat{y}|D, \hat{x}) = \frac{1}{2}\log(2\pi\hat{\sigma}^2) + \frac{(y - \hat{y})^2}{2\hat{\sigma}^2} \tag{13}$$

where $D$ represents the observation pairs $(x_{obs}, y_{obs})$, and $\hat{x}, \hat{y}, y, \hat{\sigma}^2$ denote the inquired new positions, test data, predicted mean and predicted variance, respectively. We collect a test data contains 3000 true posterior samples for averaging out SMSE the MSLL. As depicted in Fig.10 and Table 3, OPFLOW generates realistic posterior samples and achieves lower SMSE and MSLL, demonstrating superior performance over the current state-of-the-art Deep GPs.

Table 3: Comparison with benchmark GPs

| Models | SMSE | MSLL |
|---|---|---|
| OPFLOW | **0.083** | **-0.879** |
| Deep GP | 0.084 | -0.442 |
| DSPP | 0.094 | -0.673 |

### A.5 Valid stochastic processes induced by OpFlow via Kolmogorov extension theorem

In finite dimensional spaces, normalizing flow is provided to have universal representation for any data distribution under some mild conditions (Kong & Chaudhuri, 2020; Papamakarios et al., 2021). Following the fact that universal approximation of normalizing flow, we expect such universal approximation also be generalizable to OPFLOW, similar to its finite-dimensional case. However, providing a rigorous proof of

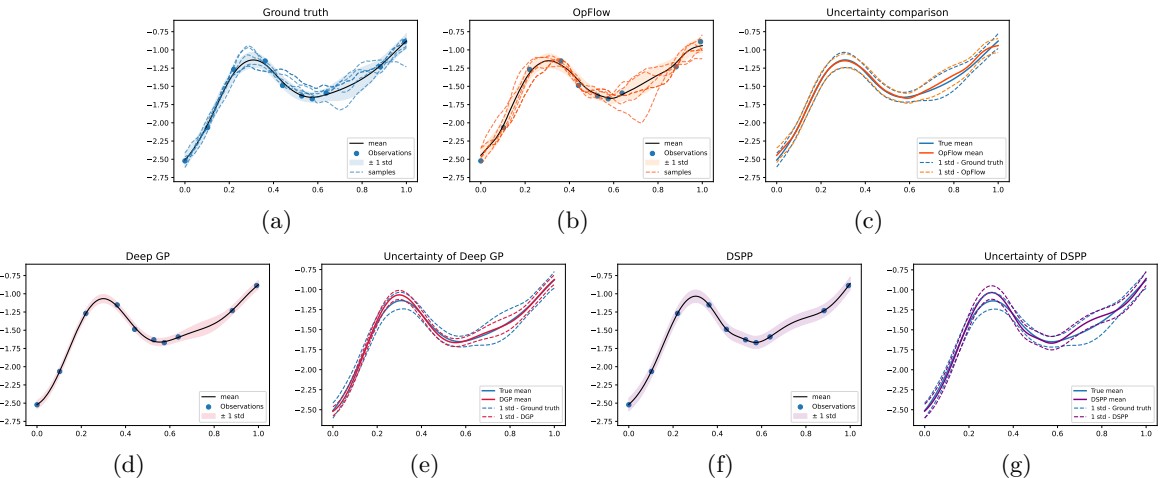

Figure 10: OPFLOW regression on GP data. (a) Ground truth GP regression with observed data and predicted samples (b) OPFLOW regression with observed data and predicted samples. (e) Uncertainty comparison between true GP and OPFLOW predictions. (d) Deep GP regression with observed data and predicted samples. (e) Uncertainty comparison between true GP and Deep GP predictions. (f) DSSP regression with observed data and predicted samples. (g) Uncertainty comparison between true GP and DSSP predictions.

universal approximation of OPFLOW remains beyond the scope of this paper and we leave it for future work. In this paper, we focus on demonstrating that OPFLOW can induce valid stochastic processes and able to learn more general stochastic processes compared to the marginal flow described in Transformed GP. The marginal flow is based on pointwise transformation of GP values, ergo, a pointwise operator. It induces valid stochastic processes and since it is limited to only pointwise transformation of GP values, it produces transformations with diagonal Jacobians (Maroñas et al., 2021). In contrast, OPFLOW acts as a more general bijective operator that results in transformations with triangular Jacobian matrix (non-diagonal) and jointly transforms the GP values. We next formally define the problem setting and provide a proof of our statements.

Consider a stochastic process $F_0 : \mathcal{X}_0 \to \mathcal{Y}_0$. For a finite sequence $x_0^1 : x_0^n = (x_0^1, x_0^2, \cdots, x_0^n)$ with $x_0^i \in \mathcal{X}_0$, we define the marginal joint distribution over the function values $y_0^1 : y_0^n = (F_0(x_0^1), \cdots, F_0(x_0^n))$. Within the framework of OPFLOW, $F_0$ is a Gaussian Process and the joint distribution $p(y_0^1 : y_0^n) = p_{x_0^1 : x_0^n}(y_0^1 : y_0^n)$ is a multivariate Gaussian. With a bijective operator $\mathcal{G}_\theta$, we have $G_\theta(y_0^1 : y_0^n) = y^1 : y^n$ and $G_\theta^{-1}(y^1 : y^n) = (y_0^1 : y_0^n)$, where $y^1 : y^n$ is defined over $x^1 : x^n$. The point alignment between $x^1 : x^n$ and $x_0^1 : x_0^n$ is determined through an underlying homeomorphic map. According to the Kolmogorov Extension Theorem (Kolmogorov & Bharucha-Reid, 2018), to establish that a valid stochastic process $F$ has $p(y^1 : y^n)$ as its finite dimensional distributions, it is essential to demonstrate that such a joint distribution satisfies the following two consistency properties:

**Permutation invariance.** Given a permutation $\pi$ of $\{1, \cdots, n\}$, the joint distribution should remain invariant when elements of $\{x^1, \cdots, x^n\}$ are permuted, such that

$$p_{x^1 : x^n}(y^1 : y^n) = p_{x^{\pi(1)} : x^{\pi(n)}}(y^{\pi(1)} : y^{\pi(n)}) \tag{14}$$

**Consistency in marginalization.** This property states that if a part of the sequence is marginalized out, the marginal distribution remains consistent with the distribution defined on the original sequence, such that for $m \geq 1$

$$p(y^1 : y^n) = \int p(y^1 : y^{n+m}) dy^{n+1:n+m} \tag{15}$$

Let's first check the permutation invariance condition. The joint distribution can be expanded as follows:

$$p(y^1 : y^n) = p(y_0^1 : y_0^n)|\det(\frac{\partial(y_0^1 : y_0^n)}{\partial(y^1 : y^n)})| \tag{16}$$

Given that $\mathcal{G}_\theta$ consists of $t$ invertible layers, we have $\mathcal{G}_\theta = \mathcal{G}_\theta^t \circ \cdots \mathcal{G}_\theta^2 \circ \mathcal{G}_\theta^1$ that gradually transforms $(y_0^1 : y_0^n)$ to $(y_1^1 : y_1^n), (y_2^1 : y_2^n) \cdots (y^1 : y^n)$. Since OPFLOW generalize RealNVP structure, the Jacobian matrix for transformations between any two adjacent layers is triangular, which implies the determination of the Jacobian matrix is the product of its diagonal components (Eq. 8. of (Kingma & Dhariwal, 2018)), then $|\det(\frac{\partial(y_i^1:y_i^n)}{\partial(y_{i+1}^1:y_{i+1}^n)})| = \Pi_{j=1}^n |\frac{dy_i^j}{dy_{i+1}^j}|$ for any $i \in \{0, \cdots, t-1\}$, and $|\det(\frac{\partial(y_0^1:y_0^n)}{\partial(y^1:y^n)})| = \Pi_{j=1}^n |\frac{dy_0^j}{dy^j}|$. Thus Eq. 16 can be simplified as:

$$p(y^1 : y^n) = p(y_0^1 : y_0^n)\Pi_{j=1}^n |\frac{dy_0^j}{dy^j}| \tag{17}$$

For this analysis, we adopt the term "volume" to refer to the determinant of Jacobian matrix discussed in in Eq. 16 or 17, align with the terminology used in RealNVP (Dinh et al., 2017). From Eq.17, we have the joint distribution under permutation $\pi$ as $p(y^{\pi(1)} : y^{\pi(n)}) = p(y_0^{\pi(1)} : y_0^{\pi(k)})\Pi_{j=1}^n |\frac{dy_0^{\pi(j)}}{dy^{\pi(j)}}|$. Given that $p(y_0^1 : y_0^k)$ is a multivariate Gaussian, which is inherently permutation invariant, the volume $\Pi_{j=1}^n |\frac{dy_0^{\pi(j)}}{dy^{\pi(j)}}|$ is independent of the order of variables, which is also permutation invariant. Thus, we have:

$$p(y^{\pi(1)} : y^{\pi(n)}) = p(y_0^{\pi(1)} : y_0^{\pi(k)})\Pi_{j=1}^n |\frac{dy_0^{\pi(j)}}{dy^{\pi(j)}}| = p(y_0^1 : y_0^n)\Pi_{j=1}^n |\frac{dy_0^j}{dy^j}| = p(y^1 : y^n) \tag{18}$$

In this way, we verify the permutation invariant property. Intuitively, this can be explained by the fact that the joint distribution $p(y^1 : y^n)$ is fully determined by a multivariate Gaussian and the volume. Under permutation of variables, the both the multivariate Gaussian and volume remain unchanged.

Now, let's verify the consistency in the marginalization. According to Eq.17, the product $\Pi_{j=1}^{n+m} |\frac{dy_0^j}{dy^j}|$ is separable and since $p(y_0^1 : y_0^{m+n})$ is a multivariate Gaussian determined from a Gaussian Process, the marginalization as specified in Eq.15 naturally holds, which verifies valid stochastic process can be induced from OPFLOW. Last, the Jacobian matrix involved in OPFLOW is a triangular matrix, which allows OPFLOW to model more general stochastic processes compared to the marginal flow used in (Maroñas et al., 2021),where the Jacobian of transformation is diagonal matrix.

### A.6 OpFlow regression on Gaussian random fields example

This is the second Gaussian random field regression example mentioned in Section 5.1. As depicted in Fig. 11, OPFLOW faithfully approximates the true posterior given the observations are concentrated along linear patterns. The relative error is the error normalized by the absolute max value of ground truth mean from GP regression.

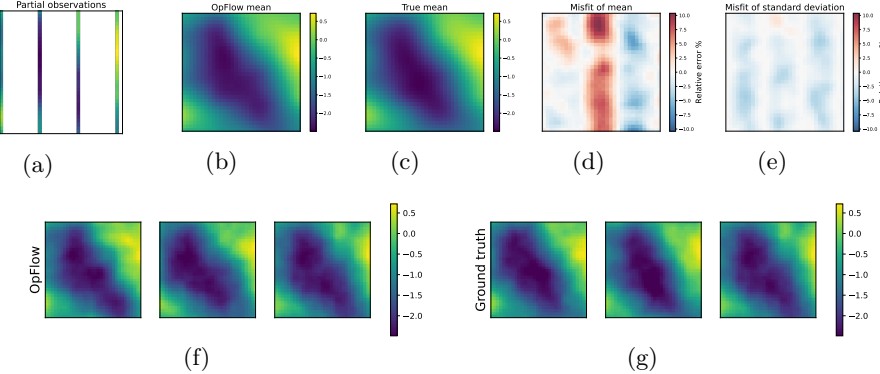

Figure 11: OPFLOW regression on 32×32 GRF data. (a) 4 strips observations. (b) Predicted mean from OPFLOW. (c) Ground truth mean from GP regression. (d) Misfit of the predicted mean. (e) Misfit of predicted uncertainty. (f) Predicted samples from OPFLOW. (g) Predicted samples from GP regression.

### A.7 More generation experiments with domain decomposed OpFlow

**GRF data.** For both OpFlow and GANO, we take the hyperparameters to be $l_{\mathcal{U}} = 0.5, \nu_{\mathcal{U}} = 1.5$ for the observed space and $l_{\mathcal{A}} = 0.1, \nu_{\mathcal{A}} = 0.5$ for the GP defined on latent Gaussian space $\mathcal{A}$. A resolution of $64 \times 64$ is chosen. Fig. 12 illustrates that OpFlow successfully generates realistic samples, closely resembling the ground truth in both autocovariance and amplitude distribution.

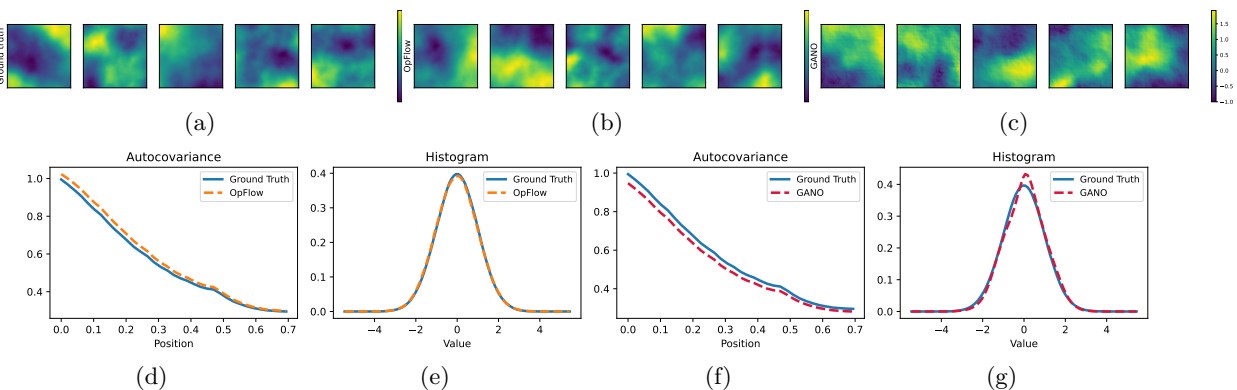

Figure 12: OpFlow generation performance on GRF data. (a) Samples from the ground truth. (b) Samples from OpFlow. (c) Samples from GANO. (d) Autocovariance of samples from OpFlow. (e) Histogram of samples from OpFlow. (f) Autocovariance of samples from GANO. (g) Histogram of samples from GANO.

**TGRF data.** We take the hyperparameters $l_{\mathcal{U}} = 0.5, \nu_{\mathcal{U}} = 1.5; l_{\mathcal{A}} = 0.1, \nu_{\mathcal{A}} = 0.5$ for generating truncated 2D Gaussian random fields. The distribution is truncated between $[-2, 2]$, with a resolution of $64 \times 64$. From Figure 13, both OpFlow and GANO generate high quality samples. In Fig. 13, we also include the histogram of the GP training data before truncation, which shows that the TGRF data strongly deviates from Gaussian. While GANO exhibits some artifacts in the generated samples, it is worth noting the covariance function and amplitude histogram are learned reasonable well, indicating that the artifacts are only present at very high frequency.

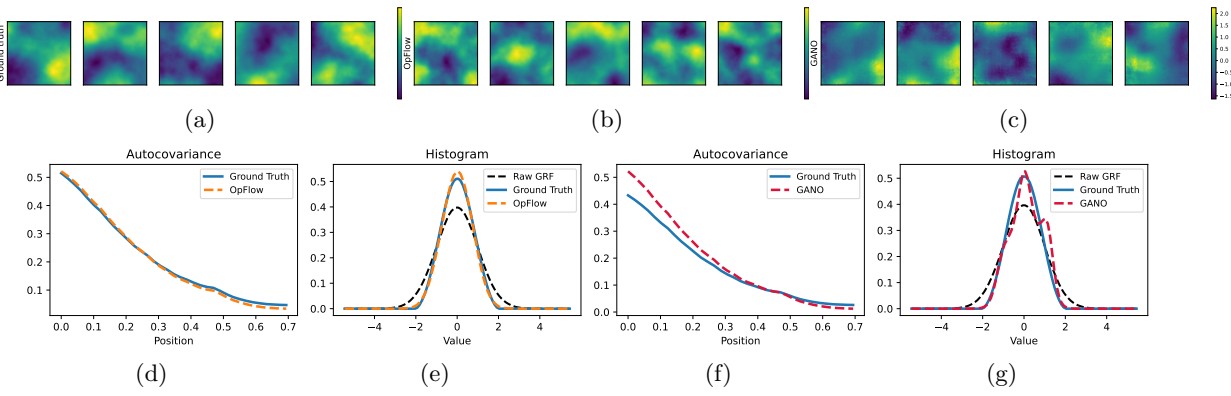

Figure 13: OpFlow generation performance on TGRF data. (a) Samples from the ground truth. (b) Samples from OpFlow. (c) Samples from GANO. (d) Autocovariance of samples from OpFlow. (e) Histogram of samples from OpFlow. (f) Autocovariance of samples from GANO. (g) Histogram of samples from GANO.

### A.8 Generation with codomain OpFlow

For the codomain implementation of OpFlow, we will focus on super-resolution tasks, which do not suffer from the masking artifacts that occur with the domain decomposed setting. It is often the case that the function of interest is multivariate, i.e. the codomain spans more than one dimension. For instance, in the study of fluid dynamics, velocity fields are often linked with pressure fields. In these cases, there exists a

natural partitioning among the co-domain for the affine coupling layers. However, in order to make codomain OPFLOW applicable to univariate functions, we first randomly shuffle the training dataset $\{u_j|_D\}_{j=1}^m$, and partition the data into two equal parts $\{u_i|_D\}_{i=1}^{m_1}$ and $\{u_i'|_D\}_{i=1}^{m_1}$. We then concatenate these parts along the channel dimension to create a paired bivariate dataset for training, $\{(u_i, u_i')|_D\}_{i=1}^{m_1}$.

**Codomain GP data.** We found that the codomain OPFLOW requires different hyperparameters from the domain decomposed setting. Here we set $l_{\mathcal{U}} = 0.5, \nu_{\mathcal{U}} = 0.5; l_{\mathcal{A}} = 0.1, \nu_{\mathcal{A}} = 0.5$, with a resolution of 256. In particular, these values suggest that codomain OPFLOW has less expressivity than domain decomposed OPFLOW, which we attribute to the need for codomain OPFLOW to learn the joint probability measures, which is hard by nature. Given that both channels convey identical information, we present results from only the first channel for samples generated by codomain OPFLOW. Fig. 14 demonstrates that both codomain OPFLOW and GANO effectively produce realistic samples, capturing both the autocovariance function and the histogram distribution accurately. Contrastingly, in the appendix, codomain OPFLOW shows great performance in super-resolution task, potentially due to the channel-split operation preserving the physical domain's integrity. To the best of our knowledge, codomain OPFLOW is the inaugural model within the normalizing flow framework to achieve true resolution invariance.

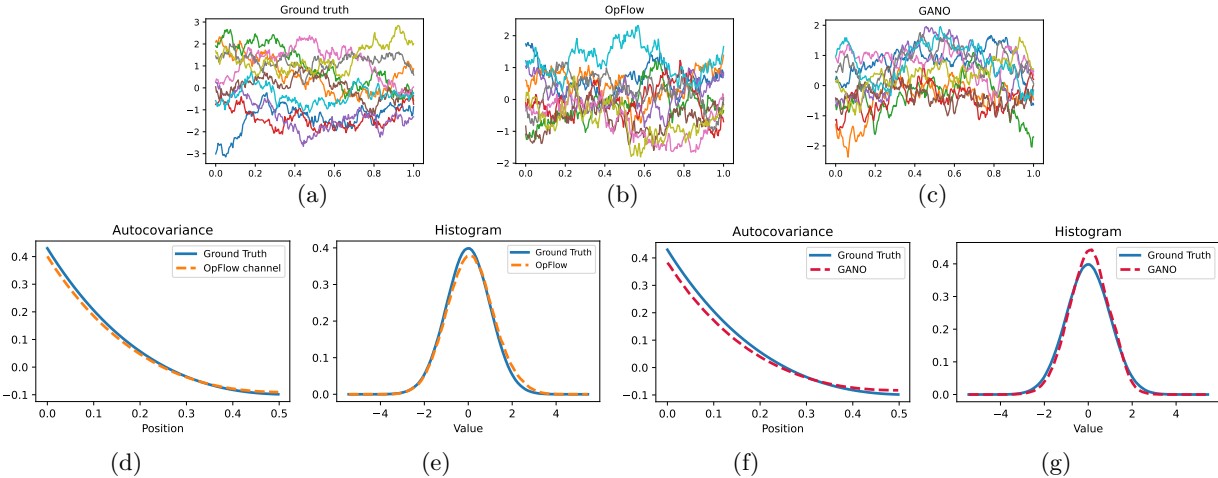

Figure 14: Codomain OPFLOW generation performance on GP data. (a) Samples from the ground truth. (b) Samples from OPFLOW. (c) Samples from GANO. (d) Autocovariance of samples from OPFLOW. (e) Histogram of samples from OPFLOW. (f) Autocovariance of samples from GANO. (g) Histogram of samples from GANO.

**Codmain TGRF data.** We set the hyperparameters to $l_{\mathcal{U}} = 0.5, \nu_{\mathcal{U}} = 1.5$ and $l_{\mathcal{A}} = 0.2, \nu_{\mathcal{A}} = 1.5$, applying bounds of -2 and 2 to the data, with a resolution of $64 \times 64$. Fig. 15 demonstrates that, under these parameters, both OPFLOW and GANO accurately capture the autocovariance function and histogram distribution. This experiment underscores codomain OPFLOW's capability in effectively handling non-Gaussian 2D data. Additionally, super-resolution results included in the appendix reveal that codomain OPFLOW is free from artifacts during super-resolution tasks, showcasing its robustness.

### A.9 Zero-shot super-resolution generation experiments of OpFlow

Here we present zero-shot super-resolution results with domain-decomposed OPFLOW and codomain OPFLOW, corresponding to the experiments detailed in the *Generation Tasks* section. All statistics, such as the autocovariance function and the histogram of point-wise values, were calculated from 5,000 realizations. Fig. 16 depicts the domain-decomposed OPFLOW, which was trained at a resolution of 256 and then evaluated at a resolution of 512. In Fig. 16(a) and (b), the super-resolution samples from domain-decomposed OPFLOW display a jagged pattern and show high discontinuities, Additionally, Fig. 16(c) and (d) show that both the autocovariance function and the histogram function evaluated from the domain-decomposed OPFLOW diverge from the ground truths. We attribute these artifacts to the domain decomposition operation, which splits the domain into smaller patches and can result in a loss of derivative continuity at the edges of these patches.

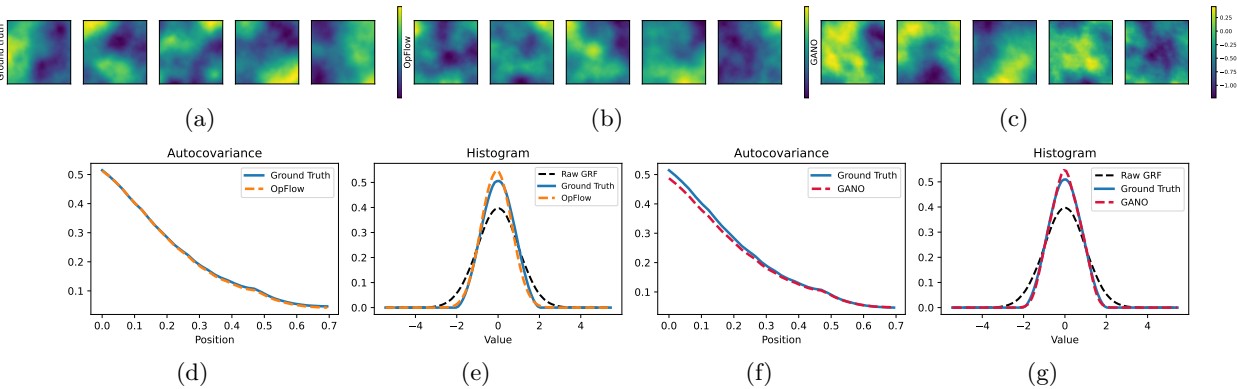

Figure 15: Codomain OPFLOW generation performance on TGRF data. (a) Samples from the ground truth. (b) Samples from OPFLOW. (c) Samples from GANO. (d) Autocovariance of samples from OPFLOW. (e) Histogram of samples from OPFLOW. (f) Autocovariance of samples from GANO. (g) Histogram of samples from GANO

This artifact is analogous to the jagged artifacts in vision transformers (ViTs), caused by the discontinuous patch-wise tokenization process (Dosovitskiy et al., 2021; Qian et al., 2021).

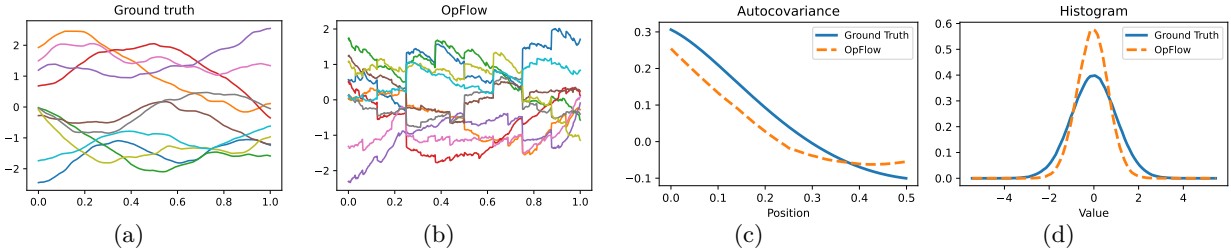

Figure 16: Domain decomposed OPFLOW super-resolution experiment on generating GP data with resolution of 512. (a) Samples from the ground truth. (b) Samples from OPFLOW. (c) Autocovariance of samples from OPFLOW. (d) Histogram of samples from OPFLOW

The super-resolution study was extended to non-Gaussian data, with results illustrated in Fig. 17. Analysis of Fig. 17 leads us to a similar conclusion that the domain-decomposition operation introduces artifacts into the super-resolution outputs of OPFLOW. This artifact is also evident in the GRF super-resolution experiment shown in Fig. 18. Here, the domain-decomposed OPFLOW was trained at a resolution of $64 \times 64$ and evaluated at $128 \times 128$, resulting in observable checkerboard patterns in the 2D super-resolution outcomes. Furthermore, the statistical analysis of the samples generated by the domain-decomposed OPFLOW, as depicted in Fig. 18, reveals a significant divergence from the ground truth.

However, in the super-resolution tasks using codomain OPFLOW for both GP and non-GP experiments, as shown in Fig. 19 and 20, the jagged or checkerboard artifacts of zero-shot super-resolution performance in OPFLOW are gone. This improvement is attributed to codomain OPFLOW maintaining the integrity of the entire input domain. Yet, this advantage does come with its own set of challenges, notably an increased difficulty in the learning process, which requires more intensive tuning efforts and may yield results that are less expressive when compared to domain-decomposed OPFLOW. Consequently, there exists a trade-off between stronger learning capability and achieving true resolution invariance when we use different partitioning methods of OPFLOW.

## A.10 Ablation study

In this ablation study, we investigate the role of 2-Wasserstein ($W_2$) regularization on the performance of the OPFLOW, as detailed in Eq. 4. We set the hyperparameters to $l_{\mathcal{U}} = 0.5, \nu_{\mathcal{U}} = 1.5$ for the observed space and

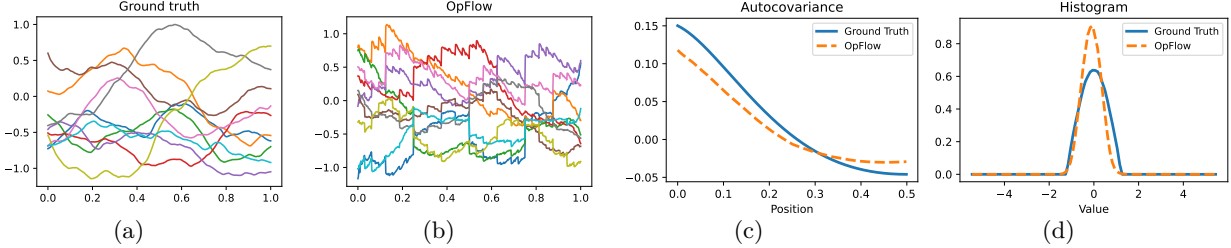

Figure 17: Domain decomposed OPFLOW super-resolution experiment on generating TGP data with resolution of 512. (a) Samples from the ground truth. (b) Samples from OPFLOW. (c) Autocovariance of samples from OPFLOW. (d) Histogram of samples from OPFLOW

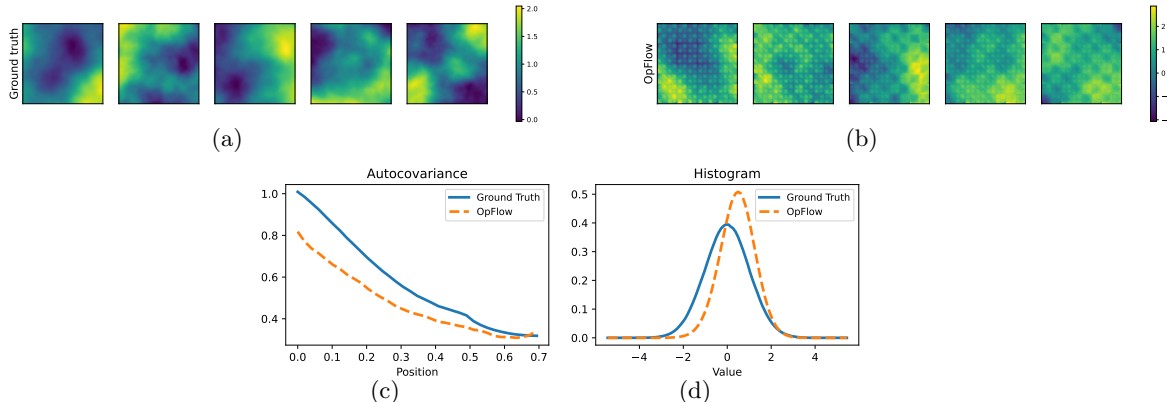

Figure 18: Domain decomposed OPFLOW super-resolution experiment on generating GRF data with resolution of 128×128. (b) Samples from groud truth. (c) Samples from OPFLOW. (d) Autocovariance of samples from OPFLOW. (e) Histogram of samples from OPFLOW

$l_{\mathcal{A}} = 0.1, \nu_{\mathcal{A}} = 0.5$ for the Gaussian Process (GP) defined in the latent space $\mathcal{A}$, with a chosen resolution of $64 \times 64$. The domain-decomposed OPFLOW model was trained under two conditions: with and without the incorporation of $W_2$ regularization during the warmup phase. According to the results depicted in Fig. 21, incorporating $W_2$ regularization significantly enhanced OPFLOW's ability to accurately learn the statistical properties of the input function data. In contrast, the model trained without $W_2$ regularization demonstrated a notable deficiency in capturing the same statistics, underscoring the critical role of $W_2$ regularization in the model's performance

### A.11 Size of training dataset for learning a prior

In this section, we revisit the 1D GP regression experiment with OPFLOW to assess the minimal dataset size required to train an effective prior. We reduced the size of the training datasets to be substantially fewer than the original count of 30000 samples. Specifically, we conducted tests using datasets of 3000, 6000, and 9000 samples to examine their impact on the regression performance.

As depicted in Fig. 22, the prior trained with 3000 samples showed reasonable performance. Although there was an observable increase in the error of the mean prediction, the uncertainty was correctly captured. This demonstrates that even a substantially reduced dataset size retains predictive utility. For the dataset of 6000 samples, we observed that the regression performance closely aligned with the ground truth and matched the performance of the prior trained with 30000 samples shown in the main text. This finding highlights that a training dataset size of 6000 is nearly as effective as one five times larger, significantly reducing data-collection demand while maintaining high fidelity in the predictions. Moreover, increasing the dataset to 9000 samples further enhanced both the accuracy and the precision of uncertainty estimation, closely approximating the performance achieved with the original dataset size of 30000.

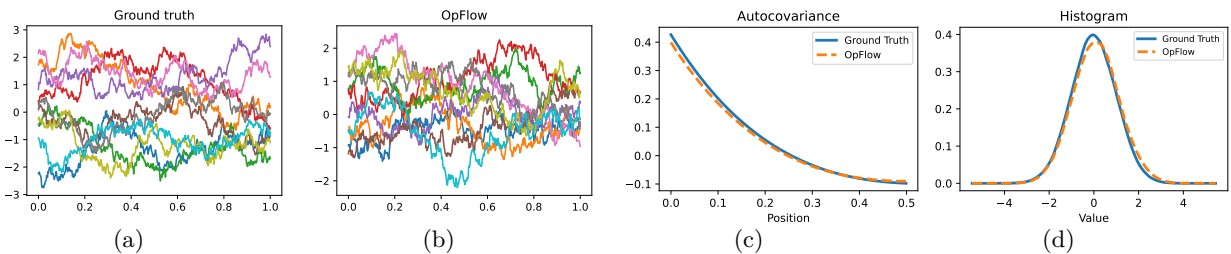

Figure 19: Codomain OpFlow super-resolution experiment on generating GP data with resolution of 512. (a) Samples from the ground truth. (b) Samples from OpFlow. (c) Autocovariance of samples from OpFlow. (d) Histogram of samples from OpFlow

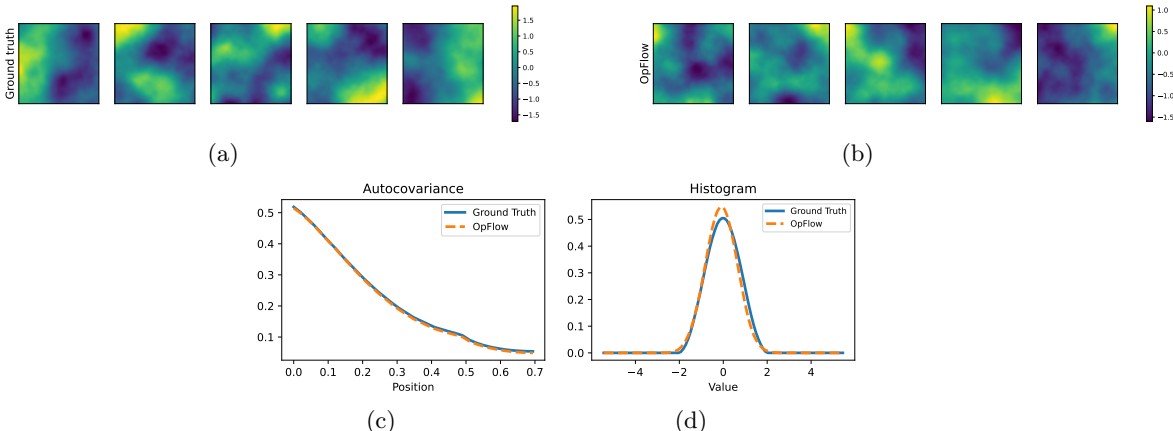

Figure 20: Codomain OpFlow super-resolution experiment on generating TGRF data with resolution of 128×128. (b) Samples from groud truth. (c) Samples from OpFlow. (d) Autocovariance of samples from OpFlow. (e) Histogram of samples from OpFlow

These results underscore the point that while larger datasets can enhance OpFlow's performance, comparable performances are achievable with much smaller datasets. In the context of GP regression with OpFlow, training datasets ranging from 3000 to 6000 samples, which is about 10% to 20% of the original training dataset size, should suffice to train a robust and effective prior. Last, it's worth exploring using an untrained OpFlow as a prior for estimating the posterior distribution of specific observations, where no training data is required. This approach is known as Deep Probabilistic Imaging (Sun & Bouman, 2020). Utilizing OpFlow in this manner may significantly broaden its applicability, particularly in scenarios where collecting training data is challenging (Aigrain & Foreman-Mackey, 2023).

### A.12 Regression with codomain OpFlow

For the training of the codomain OpFlow prior, we selected hyperparameters $l_{\mathcal{U}} = 0.5, \nu_{\mathcal{U}} = 1.5$ for the observed space and $l_{\mathcal{A}} = 0.2, \nu_{\mathcal{A}} = 1.5$ for the Gaussian Process (GP) in the latent space. In our regression task, a new realization of $u \in \mathcal{U}$ was generated, with observations taken at six randomly chosen positions within the domain $\mathcal{D}$. The goal was to infer the posterior distribution of $u$ across 128 positions. Further details on regression parameters available in Table 2. Additionally, we only show the regression results from the first channel of OpFlow generated samples. As depicted in Fig. 23, codomain OpFlow accurately captures both the mean and uncertainty of the true posterior, as well as generating realistic posterior samples that resemble true posterior samples from GPR. However, as discussed in the previous sections, codomain OpFlow is not as effective as domain-decomposed OpFlow for learning the prior, which highly affects the regression performance and may account for the underestimated uncertainty after $x = 0.5$ on the x-axis. This experiment confirms codomain OpFlow can also be used for URF, which demonstrates the robustness of UFR performance across different OpFlow architectures.

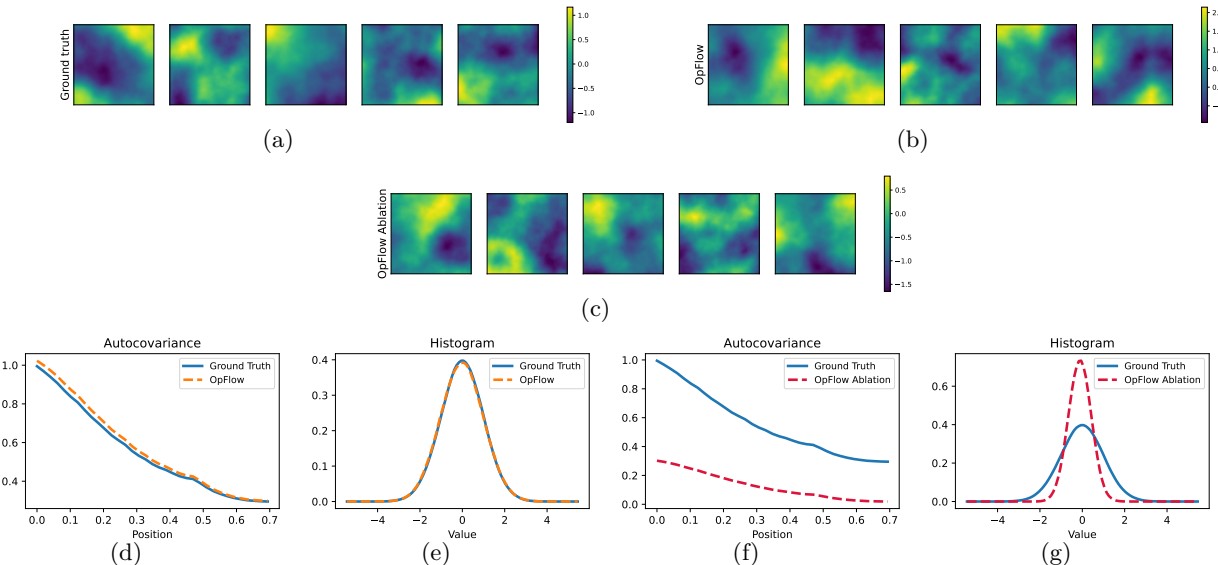

Figure 21: Ablation study of OPFLOW generation performance on GRF data (a) samples from the ground truth. (b) Samples from OPFLOW. (c) Samples from OPFLOW-ablation (OPFLOW trained without regularization). (d) Autocovariance of samples from OPFLOW. (e) Histogram of samples from OPFLOW. (f) Autocovariance of samples from OPFLOW-ablation. (g) Histogram of samples from OPFLOW-ablation

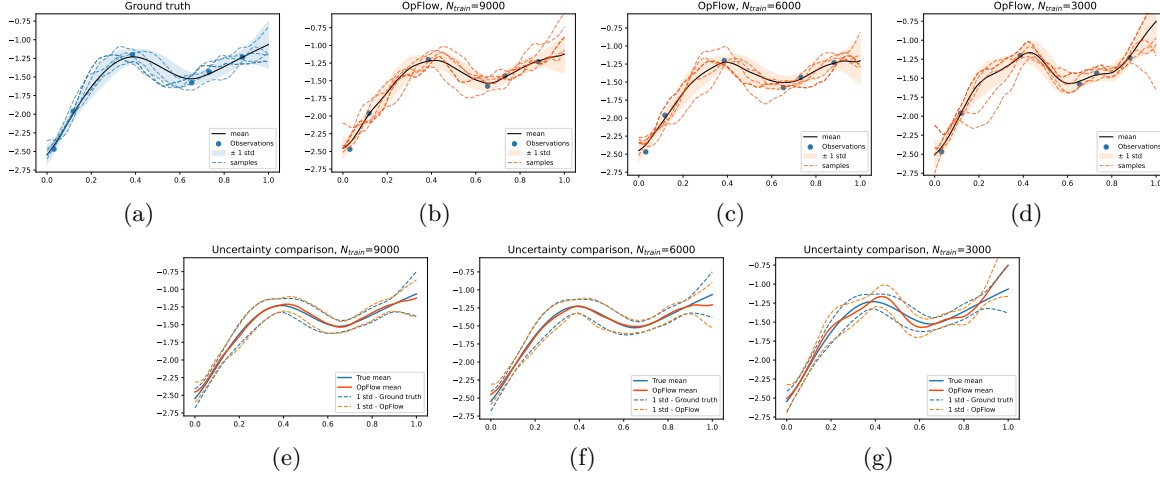

Figure 22: OPFLOW regression on GP data with varying sizes of training dataset. (a) Ground truth GP regression with observed data and predicted samples (b) OPFLOW regression with observed data and predicted samples based on the prior trained with 9000 samples. (c) OPFLOW regression with observed data and predicted samples based on the prior trained with 6000 samples. (d) OPFLOW regression with observed data and predicted samples based on the prior trained with 3000 samples. (e) Uncertainty comparison between true GP and OPFLOW predictions based on the prior trained with 9000 samples. (f) Uncertainty comparison between true GP and OPFLOW predictions based on the prior trained with 6000 samples. (g) Uncertainty comparison between true GP and OPFLOW predictions based on the prior trained with 3000 samples.

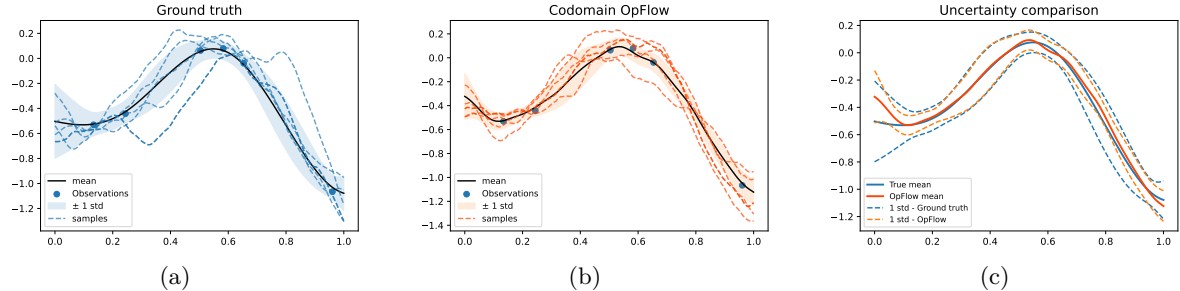

Figure 23: Codomain OPFLOW regression on GP data. (a) Ground truth GP regression with observed data and predicted samples. (b) OpFlow regression with observed data and predicted samples. (c) Uncertainty comparison between true GP and OpFlow predictions.

