# OpenReview forum: "Universal Functional Regression with Neural Operator Flows"
_TMLR — Accepted by TMLR_

### Review · Reviewer_4VjP · 2024-04-13

**Summary Of Contributions:**

The authors propose a non-Gaussian model for stochastic processes. This model a combination of both normalizing flows, which transform probability distributions in an invertible way, and neural operators, which allow to learn operators between function spaces. The theory is explained, the state of the art is explained, and various comparions to the state of the art is given in multiple experiments.

**Audience:**

Yes

**Broader Impact Concerns:**

No concerns

**Claims And Evidence:**

Yes

**Requested Changes:**

- Why do you talk about PDEs in first paragraph of introduction? I get that you are influenced by neural operators, but this mislead me when starting to read the paper.
- What does "genetic" in the footnote of page 4 mean?
- Typesetting is suboptimal. Just in equation (1), you should apply the following suggestions:
  - typeset $det$ as $\operatorname{det}$.
  - "where" in math mode can be read as the product w*h*e*r*e
  - ocassionally use \left and \right on brackets
  - Similar problems appear in many formulas
- On page 4, I do not understand the sentence "Finally, to have well-defined invertible maps between the two measures, we assume absolute continuity of measure on A and U." I think this sentence is not well-defined. Note that measures are not "absolute continuous", but "absolute continuous w.r.t. some other measure".
- Why is OpFlow discretization agnostic? I find it strange that you argue that your method is this way by just giving citations.
- Please unify the scaling of the y-axes in Figure 3.
- The colors in Figure 5 (b)&(c) are different, but seem to be the same. It is not clear to me, what colors are being used in (f) and (g). The same holds for Figure 10.
- The colors in Figure 5 (d)&(e) are very missleading. It seems that there is a high misfit, but this is hard to see. Please make the colors very clear and comment on the misfit. The same holds for Figure 10.
- Capitalization in the caption of Figure 7 (c).

**Strengths And Weaknesses:**

In general, the paper seems relevant, correct, and a good fit for TMLR. Below, there are several points that I would like to see improved or explained.

Strengths

- The idea is interesting.
- The paper is mostly well-written, with some exceptions below.
- The stochastic approaches are reasonable choices, despite my criticism in a few lines.
- The experiments are well chosen, from simple experiments giving an intuition, to real world experiments, both in regression and generative tasks.
- There are informative additional experiments and ablations in the appendix.

Weaknesses

- When you introduce your architechture in Section 4.1, I find the text somewhat vague. This is probably due to two reasons: (1) not all ingredients are my strength and (2) the connection between the ingredients is described without formulas. This prevents me from reimplementing your method. I understand that this might be too dry and long, but perhaps put this into the appendices and set a reference from 4.1 to the appendices? Similarly, in Section 4.2, I do not clearly understand how to get the inverse (but this is due to the missing details in Section 4.1).
- The loss function in (4) seems somewhat ad-hoc. I would have prefered something coming from first causes.
- The figures seem misleading, see requested changes.
- The improvements over the state of the art seem at best incremental. (This is not a problem when submitting to TMLR.)

---

> ### Author Response · Authors · 2024-04-17
> **Official Comment by Authors Part I**
>
> We appreciate the reviewer’s valuable comments and the positive take on our work. We are delighted that the reviewer found the idea interesting, the work well written, and the empirical study informative.
> In the following, we address the comments and explain changes made in the main text. These changes in the main text are colored in blue in the revised draft.
>
> **Q1. When you introduce your architechture in Section 4.1, I find the text somewhat vague. This is probably due to two reasons: (1) not all ingredients are my strength and (2) the connection between the ingredients is described without formulas. This prevents me from reimplementing your method. I understand that this might be too dry and long, but perhaps put this into the appendices and set a reference from 4.1 to the appendices? Similarly, in Section 4.2, I do not clearly understand how to get the inverse (but this is due to the missing details in Section 4.1)**.
>
> Thank you for sharing your insight about the architecture development in the subsection 4.1. We re-organized this part of the paper to mathematically define each piece of the architecture.
>
> **Q2. The loss function in (4) seems somewhat ad-hoc. I would have prefered something coming from first causes.**
>
> This is a valuable comment. At the beginning of our study, we experienced a hard time training the OpFlow model using only the loss in equation 1. We hypothesized that might be due to architecture expressiveness and typical hardness of training invertible deep learning models. Given the prior success of the GANO model, we attempted to train our architecture in the GANO framework. This attempt showed that GANO loss (infinite-dimensional Wasserstein loss) can considerably help our model training procedure. In the text, we omitted our journey behind this choice. Thanks to the reviewer’s feedback, we see the importance of including this piece of information in the main text, and made corresponding changes in Section 4.2
>
> **Q3. The improvements over the state of the art seem at best incremental. (This is not a problem when submitting to TMLR.)**
>
> We assume the reviewer is referring to improvement in the generation part rather than the regression part (please correct us if we misunderstood the reviewer’s point). If that is the case, we agree with the reviewer. The current work does not provide a noticeable advantage compared to prior works in terms of generation capabilities. Please note that generation is not the main focus of this work. The strength and focus of this work are mainly on the regression task. There is still a long way to go to make invertible models performant in generation tasks. We appreciate the comment. Please inform us if there is a need for further clarification in the text.
>
> **Q4.  Why do you talk about PDEs in first paragraph of introduction? I get that you are influenced by neural operators, but this mislead me when starting to read the paper.**
>
> We used PDEs to motivate that there are many problems in science and engineering for which we deal with function valued data. We changed the first paragraph to the following (removed the reference to PDEs),
>
> “The notion of inference on function spaces is essential to the physical sciences and engineering. In particular, it is often desirable to infer the values of a function everywhere in a physical domain given a sparse number of observation points. There are numerous types of problems in which functional regression plays an important role, such as inverse problems, time series forecasting, and data imputation/assimilation. Functional regression problems can be particularly challenging for real-world datasets because the underlying stochastic process is often unknown. “
>
> Please let us know if you suggest alternative wording.

---

> ### Author Response · Authors · 2024-04-17
> **Official Comment by Authors Part II**
>
> **Q5. What does "genetic" in the footnote of page 4 mean?**
>
> Great comment. We removed the term "generic." The only condition required is that there is a regular homomorphic map between the two domains. For example, one domain can be square, and another can be rectangular (the domains don't need to be identical). It is worth noting that $\mathcal{D_A}$ is the designer's choice, and it often makes sense to choose $\mathcal{D_A}$ identical to $\mathcal{D_U}$, unless otherwise.
>
> **Q6. Typesetting is suboptimal. Just in equation (1), you should apply the following suggestions:**
>
> We addressed the comments regarding the typeset.
>
> **Q7. On page 4, I do not understand the sentence "Finally, to have well-defined invertible maps between the two measures, we assume absolute continuity of measure on A and U." I think this sentence is not well-defined. Note that measures are not "absolute continuous", but "absolute continuous w.r.t. some other measure".**
>
> We thank the reviewer for this valuable comment. For the invertible map between two measures to be well defined, we need each measure to be absolutely continuous with respect to the other. This is achieved by adding a small amount of Gaussian noise to the data, which is also a common practice in normalizing flows, flow matching, and diffusion models. Thanks to the reviewers' comments, we updated the text to mention that they are absolutely continuous with respect to each other.
>
> **Q8. Why is OpFlow discretization agnostic? I find it strange that you argue that your method is this way by just giving citations.**
>
> The model used in OpFlow is a neural operator, and by construction, it is discretization agnostic. Following the reviewer's prior suggestion for section 4.1, we expanded on the definition of neural operators and elaborated on their discretization-invariant properties in Appendix A.2. Moreover, to support this statement, we conducted experimental studies in which we trained the model on lower resolutions in section 5.2 and tested them on higher resolutions (zero-shot super-resolution) in Appendix A.7. Furthermore, the formulation of OpFlow is for function spaces and discretization agnostic. The likelihood computation in Eq1 can take place for any discretization, and the W2 distance in Eq2 is derived for function spaces, which we show how to compute on given discretizations in Eq7.
>
> **Q9. Please unify the scaling of the y-axes in Figure 3.**
>
> Thank you for your suggestion; we have addressed this comment in the revised manuscript.
>
> **Q10. The colors in Figure 5 (b)&(c) are different, but seem to be the same. It is not clear to me, what colors are being used in (f) and (g). The same holds for Figure 10.**
>
> Thanks for pointing out this issue, we addressed this by adjusting the size of subplots and using the same scaling of y-axes for subplots in Figure 5&10.
>
> **Q11. The colors in Figure 5 (d)&(e) are very missleading. It seems that there is a high misfit, but this is hard to see. Please make the colors very clear and comment on the misfit. The same holds for Figure 10.**
>
> This is a great comment, we adjust the size of subplots to make them clear and use relative error for Figure 5 (d)&(e), Figure 10 (d)&(e). The relative error is the misfit normalized by the absolute max value of ground truth mean from GP regression. To be specific, denote $y_{predict}^{mean}, y_{predict}^{std}$ as the mean and standard deviation of predictions from OpFlow; denote $y_{gp}^{mean}, y_{gp}^{std}$ as the mean and standard deviation of the predictions from the GP (ground truth). Then relative errors used in Figure 5(d) & (e) are $(y_{predict}^{mean} - y_{gp}^{mean}) / max(abs(y_{gp}^{mean}))$ and $(y_{predict}^{std} - y_{gp}^{std}) / max(abs(y_{gp}^{mean}))$. The same for Figure 10(d)&(e). We should notice we cannot define the relative error by $(y_{predict}^{mean} - y_{gp}^{mean}) / (y_{gp}^{mean})$ because $y_{gp}^{man}$ is not a scalar, it is a 2D matrix with value ranges from -2 to 0.5
>
> **Q12. Capitalization in the caption of Figure 7 (c).**
>
> Thank you for your suggestion; we have addressed this comment in the revised manuscript

---

> ### Comment · Reviewer_4VjP · 2024-04-29
> **Answer to the rebuttal**
>
> I thank the authors for their (overly?) polite reply. The points of my review are adequatly adressed.
>
> I am looking forward of seeing how the (much better and more detailed) other reviews are being adressed, since there is no rebuttal yet.

---

> > ### Author Response · Authors · 2024-04-29
> >
> > Dear 4VjP,
> >
> > Thank you for your prompt response. Your insightful feedback has been rightfully valued. It has illuminated critical facets of the paper that were previously overlooked and highlighted nuances that may have been missed. By addressing these points, we have been able to enhance the quality of the paper, making it more accessible to readers from diverse communities and backgrounds. Such constructive critique underscores the significance of the review process, and we are grateful for its direct impact on our work.
> >
> > Best,
> > Authors

---

### Review · Reviewer_z1Tq · 2024-04-25

**Summary Of Contributions:**

The paper proposes normalizing flows for random functions, which I think reduces to warping GPs with normalizing flows.

**Audience:**

No

**Broader Impact Concerns:**

No issues

**Claims And Evidence:**

No

**Requested Changes:**

I see multiple issues with the paper.

1. The presentation is abstract and large part of the method is undefined. The paper needs to be revised extensively to make the method description self-contained and rigorous and reproducible.
2. The paper needs to more clearly position itself to the wider GP and regression literature, and demonstrate what do we gain by framing things in the (rather elegant) functional perspective wrt more traditional ML/GP style, and discuss and acknowledge deep GP literature
3. The paper needs to make its task, open problem, novelties and claims explicit and precise
4. Depending on what the claims are (not sure at the moment), the experiments need to convincingly demonstrate those. This warrants comparing against other flow/deep GPs and likely including standard GP benchmarks, and comparing against spatial GPs as well.
5. The paper needs to argue explicitly why the contributions are interesting and novel for the research community. Currently these are vague (both empirical and theoretical)

Below are more precise change requests. I’m looking forward to seeing a major revision based on these.

- The problem, task or setting of this paper is poorly described, and needs to be made explicit. The paper seems to coin a new concept of UFR, but doesn’t define what it is, and only vaguely describes it. I’m quite confused by this: I’ve not heard of UFR before and I don’t see how this differs from just regression. It’s not clear what is the task in this paper (GP regression?), what is the open problem (non-shallow GPs?).
- The paper presents its material from a functional viewpoint, which is elegant but it’s not made clear how does this differ from the traditional ML approach. Regular GPs (and even neural networks) are all functions as well, or are they not? What do we gain from framing everything in terms of function spaces? The introduction claims that earlier methods are lacking function space priors or likelihoods. Err… what? This is obviously false. Please contextualise and contrast the paper’s ideas with more conventional ML approaches. The paper needs to clearly explain why it chooses the functional perspective, and what do you gain by doing so, and how it differs from conventional perspective.
- The claims of this work are ambiguous and need to be made precise. The intro says that they develop a “viable” formulation. What does this mean? Didn’t we have any “viable” regression functions before? We certainly do! Again, I don’t understand where this paper is coming with statements like this. The paper needs to make its claims precise and contextualise them to the earlier works. The open problem, contributions, claims all need to be rigourous and contextualised to earlier works.
- I think ultimately the contribution is a GP warped by a normalizing flow. This is fantastic idea! However, it already exists by Maronas et al (2020) and Klein et al (2022) and also arguably by Hegde et al (2019) and Ustyuzhaninov et al (2020). The paper needs to cite, discuss, contrast, and compare to these methods. It also needs to contrast and compare to deep GPs. There are also deep kernel methods (DKM, DKP), and spatial GPs (eg. GPRN) that seem to tackle the same problems (and thus warrant comparison or discussion).
- Overall this paper is not self-standing: the task and model have not been defined, and perhaps these definitions are in some other papers (but none are cited for the “UFR”...). The Sec 3 does not define the task, or relate the math to GP regression. I don’t understand what the {u,a,F,G} are. I don’t understand why don’t we want to find a=u, but instead we want to find some other function a and then make it only match data through some “forward” (not explained either). The paper then says that F=G=I so nothing happens. Err… what? Section 4 seems misplaced, since it starts with some informal and vague description of the implemention details before the model has been defined in any way! Please define the task, the model and the inference all in rigorous math that can be understood without reading other papers first.
- The eq 1 describes the normalizing flow, but I’m confused where does this come from or what is it. We already know normalizing flows, so is this just a standard NF or something different? This needs to be exposed better. The eq 1 is also quite casually presented, and needs more rigor on its derivation and showing its a valid object. It would also be good to somehow illustrate what happens here, and what each layer is doing to the base GP.
- I can’t identity the motivation or justification for the Wasserstein stuff. The Wasserstein seems to do the same thing as the flow, so now we have two losses that both match densities through a transport. Isn't this redundant. The Wasserstein is also poorly motivated, since it’s not clear what problem it fixes (apart from normalizing flow being “insufficient”).
- In eq 3 it is assumed that FU is a GP. This is a drastic assumption, and this is surely almost surely never true. If the F can map U to a GP, it probably already perfectly matches to A as well, and then we have already succeeded. Generally the FU is not a GP, and you can’t do this. Can you comment on this, and discuss what repercussions this has? I also don’t understand where the kernels K_1 and K_2 come from. Surely you can’t know the kernel of FU in any way.
- I can’t follow eq 5. We have in left (i,j), in middle the j disappears, in right the j appears again. Can you explain or fix?
- I don’t understand where the m’s and K’s are coming from.
- Alg 1 does a bunch fo stuff that is not described in the paper. Everything that happens in alg1 needs to be properly explained in main text.
- I don’t understand the phi stuff. Not sure what is u_phi, or what is a_phi or F here. I can’t map this notation to the regression problem or to inputs or outputs. I don’t understand where the F appears here, or what does it have to do with regression. Didn’t we assume F=I? Didn’t the eq 1 already solve the regression problem? Why are we doing some extra stuff in addition to solving eq 1? I’m quite confused of the wider setting or scheme that is implicitly assumed in this paper. Can you help me understand? Usually regression approach start by defining y = f + eps, and then state likelihood p(y|f) and prior (f) and state the problem as finding p(f|y). Can you start from the beginning and explain what you do in this paper step by step?
- Eq 8-10 seems to be some kind fo redefinition of basic GP equations. Can you relate them back to the conventional equations, and describe how this is different (or beneficial). Why not go for the conventional notation?
- I’m confused of alg2: isn’t this just standard NF? Are you describing something novel or known?
- How do you compute the determinant? What kind of NF is this (discrete, continuous, planar, sylvester, etc)?
- At experimental section I’m still not sure what UFR means, so I’m not sure what claims are you trying to demonstrate. To me this just seems GP regression and GP density fitting using output-warped GPs. In that case you need to compare to competing GP methods that do similar (ie. warped GPs, deep GPs, flow GPs). There’s also deep kernel learning and deep kernel processes that should probably be acknowledged or compared against.
- If the paper claims to develop more flexible deep GPs, then it needs to demonstrate these on standard GP benchmarks.
- Throughout the paper it's unclear what parts are novel and what known. Please clarify throughout.

**Strengths And Weaknesses:**

S: The idea of normalizing flow GP is very strong, and this model family has great potential.

S: The writing is good and flows well. With below changes and polishing this has potential to become a nice paper.

W: I’m not sure if this idea is novel

W: The presentation is incomplete and insufficiently rigorous, and seems to detach itself from the conventional GP notation and community making it hard to understand how this positions in the GP literature, or where should it position itself.

---

> ### Author Response · Authors · 2024-05-06
> **Official Comment by Authors Part I**
>
> We appreciate the reviewer’s comments. We are delighted that the reviewer found the idea of extending normalizing flow to function space very strong, and our writing and the flow of the paper clear. It’s worth noticing that our work introduces an infinite-dimensional extension of normalizing flow, termed OpFlow, which is designed specifically for both regression and generation tasks. OpFlow learns mapping between infinite-dimensional function spaces, which enables non-diagonal transformations while ensuring that output of each layer of OpFlow is a valid stochastic process. We provided a detailed explanation of OpFlow's resolution-agnostic property in Appendix A.2 and zero-shot super-resolution experiments in Appendix A.7.
>
> Before the point-by-point reply, we first provide the following generic explanation. The setting of the paper is as follows,
>
> 1- Given many functions, learn the prior over the function space. Please note that, data samples here are functions, not each point evaluation of the functions. The learning with OpFlow enables non-diagonal transformations and can be used for modeling arbitrary copula.
>
> 2- Use the learned prior to do GP-style regression with accurate uncertainty quantification that doesn’t require variational inference.  Since this is no longer a GP regression, we coined a term UFR for this task, which is explained in section 3.
>
> To give more intuition, let's see how we traditionally used to tackle this problem of regression on function spaces. (the step 1 above): A person had to look at the training data set of functions, manually decide what GP kernel fits the data (this is the kernel engineering step in the traditional method) and (step 2 above), then apply GP regression using the fitted GP on a new point evaluations of the test function. As the reviewer may see, this traditional approach makes a strong assumption that the data prior is still Gaussian, and therefore, the regression can be carried out using GP regression, a limitation that our work attempts to resolve. Please note that the GP regression setting is slightly different from that of the generic deep GP. In deep GP, we fit a GP to the whole dataset, meaning each data point is considered a point evaluation of a function. Please let us know if we can discuss this further.
>
> In the following, we address the comments and explain changes made in the main text. These changes in the main text are colored in blue in the revised draft.
>
> **Q1. The problem, task or setting of this paper is poorly described, and needs to be made explicit. The paper seems to coin a new concept of UFR, but doesn’t define what it is, and only vaguely describes it. I’m quite confused by this: I’ve not heard of UFR before and I don’t see how this differs from just regression. It’s not clear what is the task in this paper (GP regression?), what is the open problem (non-shallow GPs?).**
>
> We added a section at the beginning of section 3, termed as “Universal functional regression”. In this part, we provide a dedicated section to UFR and explain the setting.
>
> **Q2. The paper presents its material from a functional viewpoint, which is elegant but it’s not made clear how does this differ from the traditional ML approach. Regular GPs (and even neural networks) are all functions as well, or are they not? What do we gain from framing everything in terms of function spaces? The introduction claims that earlier methods are lacking function space priors or likelihoods. Err… what? This is obviously false. Please contextualise and contrast the paper’s ideas with more conventional ML approaches. The paper needs to clearly explain why it chooses the functional perspective, and what do you gain by doing so, and how it differs from conventional perspective.**
>
> As explained above, OpFlow is an infinite-dimensional normalizing flow, which is  able to model arbitrary copula with non-diagonal transformation while ensuring the output of each layer is a valid stochastic process. This work can be considered as an extension of GP regression where instead of considering Gaussian prior assumption, the prior is learned directly from functions in the data set . The motivation behind function view is identical to the motivation of developing GP in the first place, which is to deal with point evaluation of function. Earlier generative methods mentioned in the introduction section indeed lack likelihood estimation. You can not use GANO (Rahman et al ,2022a), VANO (Seidman et al 2023), or DDO  (Lim et al, 2023) or their finite-dimensional counterparts (GAN, VAE, Diffusion Model) to provide precise likelihood estimation.

---

> > ### Author Response · Authors · 2024-05-06
> > **Official Comment by Authors Part II**
> >
> > **Q3. The claims of this work are ambiguous and need to be made precise. The intro says that they develop a “viable” formulation. What does this mean? Didn’t we have any “viable” regression functions before? We certainly do! Again, I don’t understand where this paper is coming with statements like this. The paper needs to make its claims precise and contextualise them to the earlier works. The open problem, contributions, claims all need to be rigourous and contextualised to earlier works.**
> >
> > We appreciate the reviewer’s comment. We certainly appreciate it if the reviewer points us to the setting Bayesian regression on function spaces where the prior is learned from data (characterized by arbitrary copulas) instead to be assumed.
> >
> > **Q4. I think ultimately the contribution is a GP warped by a normalizing flow. This is fantastic idea! However, it already exists by Maronas et al (2020) and Klein et al (2022) and also arguably by Hegde et al (2019) and Ustyuzhaninov et al (2020). The paper needs to cite, discuss, contrast, and compare to these methods. It also needs to contrast and compare to deep GPs. There are also deep kernel methods (DKM, DKP), and spatial GPs (eg. GPRN) that seem to tackle the same problems (and thus warrant comparison or discussion).**
> >
> > We added a subsection “Deep and Warped Gaussian Processes” in Section 2 to compare our models with other models. Klein et al (2022) is an inappropriate reference, which employed RealNVP with GP for modeling laser-induced breakdown spectroscopy. However, as pointed out in Maronas et al (2020) , non-diagonal transformations achieved by normalizing flows must ensure that each layer's output remains a valid stochastic process. RealNVP, being a finite-dimensional normalizing flow, learns probability distributions rather than the probability measures on the function spaces induced by potential stochastic processes, and thus doesn't define a valid function prior. Moreover, Klein et al (2022) relies on the assumption that data reside on a lower-dimensional manifold and only enables inference on fixed-size and regular grids, which violates the infinite-dimensional nature of GP. In the end, we would be happy to contrast our work to any deep GP paper that the reviewer has in mind.
> >
> > **Q5. Overall this paper is not self-standing: the task and model have not been defined, and perhaps these definitions are in some other papers (but none are cited for the “UFR”...). The Sec 3 does not define the task, or relate the math to GP regression. I don’t understand what the {u,a,F,G} are. I don’t understand why don’t we want to find a=u, but instead we want to find some other function a and then make it only match data through some “forward” (not explained either). The paper then says that F=G=I so nothing happens. Err… what? Section 4 seems misplaced, since it starts with some informal and vague description of the implemention details before the model has been defined in any way! Please define the task, the model and the inference all in rigorous math that can be understood without reading other papers first.**
> >
> > Thanks for the input. We added the task description to the beginning of section 3. We also added details on the model description. Please refer to the blue colored changes in the paper.  We need to mention, section 3 is used to describe the math background for developing infinite-dimensional normalizing flow, not related to GP regression. The notations, such as  $\mathcal{U, A, F, G}$ and narrative language used adhere to conventions in operator learning (FNO, Li et al 2020; Neural Operator, Kovachki 2021; etc). The paper focuses on learning the prior from the data, without involving transforming likelihood of the GP output, $\mathcal{G}$ and $\mathcal{F}$ represent the forward and inverse operator, with $\mathcal{G} = \mathcal{F^{-1}}$.
> >
> > **Q6. The eq 1 describes the normalizing flow, but I’m confused where does this come from or what is it. We already know normalizing flows, so is this just a standard NF or something different? This needs to be exposed better. The eq 1 is also quite casually presented, and needs more rigor on its derivation and showing its a valid object. It would also be good to somehow illustrate what happens here, and what each layer is doing to the base GP.**
> >
> > The equation 1 is the generalization of standard normalizing flow to function spaces when the function can be presented at any discretization D. Please note that, in this setting, each data point has its own discretization D.

---

> > > ### Author Response · Authors · 2024-05-06
> > > **Official Comment by Authors Part III**
> > >
> > > **Q7. I can’t identity the motivation or justification for the Wasserstein stuff. The Wasserstein seems to do the same thing as the flow, so now we have two losses that both match densities through a transport. Isn't this redundant. The Wasserstein is also poorly motivated, since it’s not clear what problem it fixes (apart from normalizing flow being “insufficient”).**
> > >
> > > This is a valuable comment. At the beginning of our study, we experienced a hard time training the OpFlow model using only the loss in equation 1. We hypothesized that might be due to architecture expressiveness and typical hardness of training invertible deep learning models. Given the prior success of the GANO model, we attempted to train our architecture in the GANO framework. This attempt showed that GANO loss (infinite-dimensional Wasserstein loss) can considerably help our model training procedure. In the text, we omitted our journey behind this choice. Thanks to the reviewer’s feedback, we see the importance of including this piece of information in the main text, and made corresponding changes in Section 4.2
> > >
> > > **Q8. In eq 3 it is assumed that FU is a GP. This is a drastic assumption, and this is surely almost surely never true. If the F can map U to a GP, it probably already perfectly matches to A as well, and then we have already succeeded. Generally the FU is not a GP, and you can’t do this. Can you comment on this, and discuss what repercussions this has? I also don’t understand where the kernels K_1 and K_2 come from. Surely you can’t know the kernel of FU in any way.**
> > >
> > > The derivation does not assume $\mathcal{F_{\theta} \sharp U}$ is GP, it first finds a GP fit to FU and then computes the W2 between the GP fit and A GP.
> > >
> > > **Q9. I can’t follow eq 5. We have in left (i,j), in middle the j disappears, in right the j appears again. Can you explain or fix?**
> > >
> > > There is a typo, thanks for pointing it out. The $e_i$ should be $e_j$.
> > >
> > > **Q10. I don’t understand where the m’s and K’s are coming from.**
> > >
> > > The parameter m is the mean of the GP and K is the covariance operator of the GP.
> > >
> > > **Q11. Alg 1 does a bunch fo stuff that is not described in the paper. Everything that happens in alg1 needs to be properly explained in main text.**
> > >
> > > Thanks for the suggestion. We brought the explanation to the main text.
> > >
> > > **Q12. I don’t understand the phi stuff. Not sure what is u_phi, or what is a_phi or F here. I can’t map this notation to the regression problem or to inputs or outputs. I don’t understand where the F appears here, or what does it have to do with regression. Didn’t we assume F=I? Didn’t the eq 1 already solve the regression problem? Why are we doing some extra stuff in addition to solving eq 1? I’m quite confused of the wider setting or scheme that is implicitly assumed in this paper. Can you help me understand? Usually regression approach start by defining y = f + eps, and then state likelihood p(y|f) and prior (f) and state the problem as finding p(f|y). Can you start from the beginning and explain what you do in this paper step by step?**
> > >
> > > Thanks for the comment. the section 4.2 describes how to develop infinite-dimensional normalizing flow, and is not related to the GP regression. The notations we used here adhere to conventions in operator learning (FNO, Li et al 2020; Neural Operator, Kovachki 2021; etc). This paper describes how to learn a prior on general data function spaces, not involving transforming the likelihood of the outputs of GP. Lets first explain what GP regression does using the notation above. In GP regression, there observation y is equation to noisy observation of the underlying function f at a few evaluation points D. Therefore, using your notation, the model is $y = f\_{|D} + epsilon$. Then in GP regression, we ask what is $P(f\_{|D \cup D’} |y)$ for new points D’. In GP regression, the prior (f) you mentioned is assumed to be Gaussian. In our paper, it is learned from data.
> > >
> > > **Q13. Eq 8-10 seems to be some kind fo redefinition of basic GP equations. Can you relate them back to the conventional equations, and describe how this is different (or beneficial). Why not go for the conventional notation?**
> > >
> > > These equations are expansions of log probabilities using Bayes rule. The inference of OpFlow is based on Eq 8-10.
> > >
> > > **Q14. I’m confused of alg2: isn’t this just standard NF? Are you describing something novel or known?**
> > >
> > > OpFlow is not the standard finite-dimensional NF. In contrast, OpFlow is an infinite-dimensional normalizing flow, which  performs non-diagonal transformations while ensuring that output of each layer is a valid stochastic process. This algorithm shows how we can sample posterior on function spaces using Langevin dynamics and OpFlow, which is an extension of NF to function spaces.

---

> > > > ### Author Response · Authors · 2024-05-06
> > > > **Official Comment by Authors Part IV**
> > > >
> > > > **Q15. How do you compute the determinant? What kind of NF is this (discrete, continuous, planar, sylvester, etc)?**
> > > >
> > > > Thanks to the reviewers suggestion, we added the description to the paper. The forward and inverse processes as well as the calculation of the determinant are summarized in Appendix A.1. We re-emphasize that OpFlow is not the finite-dimensional normalizing flow model; the model in our work is an operator between two infinite-dimensional function spaces, inspired by the now classical RealNVP and Glow architectures (Kingma, 2018)
> > > >
> > > > **Q16. At experimental section I’m still not sure what UFR means, so I’m not sure what claims are you trying to demonstrate. To me this just seems GP regression and GP density fitting using output-warped GPs. In that case you need to compare to competing GP methods that do similar (ie. warped GPs, deep GPs, flow GPs). There’s also deep kernel learning and deep kernel processes that should probably be acknowledged or compared against.**
> > > >
> > > > We added a section at the beginning of section 3, termed as “Universal functional regression”. In this part, we provide a dedicated section to UFR and explain the setting.  We also added a section at the end of section 2, termed as “Deep and Warped Gaussian Processes” to compare OpFlow with other competing GP methods.
> > > >
> > > > **Q17. Throughout the paper it's unclear what parts are novel and what known. Please clarify throughout.**
> > > >
> > > > Our contributions are as follows:  we develop Neural Operator Flows (OpFlow), an infinite-dimensional extension of normalizing flows for both generation and regression tasks. OpFlow is an invertible operator that maps the (potentially unknown) data function space into a Gaussian process
> > > >
> > > > Compared to pioneering work of Maronas et al (2020), regression with OpFlow introduces two key enhancements: (1) OpFlow enables non-diagonal transformations while ensuring that output of each layer of OpFlow is a valid stochastic process. (2) Robust and accurate uncertainty quantification via drawing posterior samples of the Gaussian process with SGLD and subsequently mapping them into the data function space, thereby obviating the need for variational inference.

---

### Review · Reviewer_7nyT · 2024-04-26

**Summary Of Contributions:**

The authors introduce a new form of nonparametric regression where a specific form of learnable and invertible prior over functions is used for addressing functional regression. The method builds extensively on modern literature that is covered well, and the approach is demonstrated clearly in (mostly synthetic) examples tasks.

**Audience:**

Yes

**Broader Impact Concerns:**

None.

**Claims And Evidence:**

Yes

**Requested Changes:**

Practical suggestions on how to address my concern with minimal effort:
- You could add one synthetic data experiment that explicitly quantifies how much data is needed, so that you can make a clear conclusion on when the approach can be used. For example, just run the GP example with smaller and smaller training sets and show what happens. By explicitly showing that you need a certain amount of training samples to learn a useful prior you would clearly communicate the property. Alternatively, if already a few tens/hundreds of samples would turn out to be enough (for the 1D example), you would get to state that as an additional result.
- Explicitly mention in both Introduction and Conclusion that compared to the traditional GP regression you require additional data, and in case this data is not available the method is not applicable and GP -- despite its other simplifying limitations -- remains a practical tool of choice. Emphasise that assuming a specific type of stochastic process beforehand has its own merits, in particular processes like GP with very few learnable parameters and strong inductive biases (e.g. Aigrain&Foreman-Mackey explicitly discuss how physical prior knowledge is encoded into kernels). Check which of your motivational examples actually fit within your scope; I am not sure myself whether in the astronomical applications we would have the suitable training samples or not.

**Strengths And Weaknesses:**

The work builds on broad current literature, providing almost review-like coverage on recent advances in several topics in Section 2. The mathematical development appears sound and the empirical experiments demonstrate the model achieves its goals, in effect confirming the methodological development to a sufficient degree. The method works well in the experiments considered and the experiments are informative and clear, considering also the generative side is a nice addition, and the Supplement provides extensive technical details that help understanding the details. From these perspectives, the paper clearly satisfies the evaluation criteria and I would recommend acceptance.

However, I have one concern about the positioning and narrative for readers who are not inherently familiar with the concept of learnable priors and who are looking for practical tools to address the same kind of tasks they today use GPs for. I would like to see the authors make it more transparent that with the added flexibility comes quite substantial additional requirement of having relevant training data. Note that this is not a problem as such since the task is clearly interesting in the current form, I am merely asking for communicating it better. I feel the paper slightly too strongly suggests, even though implicitly, the proposed method as a better plug-in replacement for GPs and in doing so can mislead some readers, and this is getting close to the spirit of the TMLR evaluation criterion about validity of the claims. I marked that the explicit claims are supported by evidence, but from this perspective I consider it a borderline case.

In brief, your model appears to require a training data set of substantial size; for the synthetic experiments you use 30,000 examples even for the simple 1D GP and TGP cases and 20,000 for the 2D random field, and in the seismic waveform problem you assume access to 20,643 time series. While these are not massively large numbers, they are still way above what would be available in many of the use-cases where nonparametric regression building on GPs (the starting point of your story) are used. In fact, in many concrete cases there are no training instances of this kind, but rather the models need to be fit directly on the data you consider as test cases (e.g. the 60 randomly selected observation points for the new time series in the seismology case). I fully agree there are use cases where such training data can be available (like your particular historical seismic data collection), but they are not the most common ones especially in empirical sciences where the measurements are done in a laboratory.

---

> ### Author Response · Authors · 2024-05-06
> **Official Comment by Authors Part I**
>
> We appreciate the reviewer’s valuable comments and the positive take on our work. We are delighted that the reviewer found our work on  nonparametric regression valuable.
> To this point, we want to express that the reviewer made a very crucial point that we very much appreciate, and for the good of our community and this work, we will incorporate it in the paper.
>
> In the following, we address the comments and explain changes made in the main text. These changes in the main text are colored in blue in the revised draft.
>
> **Q1. You could add one synthetic data experiment that explicitly quantifies how much data is needed, so that you can make a clear conclusion on when the approach can be used. For example, just run the GP example with smaller and smaller training sets and show what happens….**
>
> This is a great suggestion, we explore the minimal size of the training dataset to train an effective prior. In Appendix A.9, we detail our analysis of the 1D GP regression experiment using OpFlow trained with reduced training dataset sizes. From this investigation, we conclude that the training dataset with 3000 to 6000 samples (10% to 20% of the original dataset size) is sufficient to train a robust and effective prior.
>
> **Q2. Explicitly mention in both Introduction and Conclusion that compared to the traditional GP regression you require additional data, and in case this data is not available the method is not applicable and GP -- despite its other simplifying limitations -- remains a practical tool of choice. Emphasise that assuming a specific type of stochastic process beforehand has its own merits, in particular processes like GP with very few learnable parameters and strong inductive biases (e.g. Aigrain&Foreman-Mackey explicitly discuss how physical prior knowledge is encoded into kernels). Check which of your motivational examples actually fit within your scope; I am not sure myself whether in the astronomical applications we would have the suitable training samples or not.**
>
> We thank the reviewer for this valuable comment. Our method is relevant only when we have access to training data to learn the prior. While in GP regression, the practitioner often optimizes the used kernel using a few data points, the plain GP does not require even that. However, GP regression without tuning is known to not be a considerably useful tool in practice, tuning it requires minimal samples. We elaborate on this point at the end of the introduction section.
>
> Last, we suggest a possible approach with OpFlow for addressing regression problems where no training data is available. This method, known as Deep Probabilistic Imaging (DPI) (Sun & Bouman, 2020), utilizes an untrained normalizing flow model as a prior to estimate the posterior distribution of an unseen image. In DPI, the weights of the normalizing flow are optimized to make the generated samples from the model match a specific observation. We may extend this framework to infinite-dimensional space with OpFlow, and further study is required.

---

### Comment · Reviewer_z1Tq · 2024-05-19

Thanks for the responses. I re-read the manuscript, and there are still significant problems in terms of presentations, with some comments below. I believe the manuscript still needs a major revision.

- Sec 3 UFR paragraph refers U to data space, but U is a function space. Usually data is vector-valued observations, not functions [unless eg. in audio]. It would be good to clarify the nature of “data” in this paper.
- In sec 3 UFR paragraph you observe data at D_j, but evaluate them at D_dagger. Similarly you observe u_j, but consider density of u. I don’t really know what is the dagger stuff, or what is u (where does it come from?). Is u some kind of test point? I don’t really follow.
- I don’t even understand the setting: I thought we were supposed to maximize the probability of the observed functions u_j, not some other mysterious function u.
- I also don’t understand what is the “p” in the max. Is this the true “p”, or some parametric “p”? I don’t really understand the problem at all. Are we trying to find some true underlying u; or maximize true probability of observations; or maximize some parametric learnt probability of observations (or true function); or maybe you want to learn a measure instead (ie. some set function \mu… from measure theory?).
- Also, the optimisation task of max_u p(u) is clearly pathological, and can be optimised by having a Dirac function on any arbitrary u \in U.
- Even if the argmax refers to the u_j, you could still have a pathological solution by just overfitting to u_j. I don’t see anything in the formulation that would ensure generalisation (or even discuss it). The “posterior distribution” is undefined, so I’m not sure what it means.
- In GPR we do not try to maximize likelihood of data. Instead, we would find the posterior, and try to learn the kernel also.
- It’s also vague what is the relationship between p and P_U. It’s also vague what is the relationship between p in the first paragraph of sec 3, and the p in the argmax. Are they the same p? I don’t think so, since the argmax-p is learnt, so surely it should contain some parameters.
- The “invertible operators” paragraph repeats math. This gives the impression that the two paragraphs in sec 3 are two separate stories. Are they? It would be good to harmonize the material in the paper.
- “a \in A is drawn… and consequently a_j is a…”. Do we sample a or a_j? Where did the a_j come from?
- How can you have an invertible mapping if A and U have different domains? Is this necessary?
- What does “small amount of GP noise” mean? What is “GP noise”? Why do you need to add extra noise, or why does it ensure continuity?
- In eq 1 the P_U is unknown. How do you sample from it? You do observe u_j|D, but this is very different from obtaining entire u_j, which is what eq 1 calls for.
- I dont’ understand how the domain points evolve in eq 1, or why is that a valid thing to do. It would be good to have an illustration of the functional flow, and how it works. I don’t really understand this yet intuitively.
- Eq 1 is still just appearing in this paper, and there is no derivation or construction of this. This is the main novelty of the paper, and it needs a strong exposition and derivation. I’m not convinced that this is a valid mathematical construct, and I suspect that the authors are casually assuming (or hoping) that standard normalizing flow can be written like this.
- I also notice that I’m now just re-writing my earlier comments. Please address all of my concerns (current and previous) directly in the manuscript.
- Last sentence of eq 1: I’m not sure I agree with this. Can you proof or show this? Why does the eq 1 “measure” distance?
- The whole point of the paper is to do mappings between functions. Yet, it seems that you immediately discretise back into standard grids of images. Doesn’t this then just reduce to a standard normalizing flow? What extra does the OpFlow bring us if we discretise anyways?
- I don’t understand where does GP_1, GP_2, K_1, K_2 come from. What are these? I struggle to follow the Wasserstein story. I struggle to understand what are we trying to achieve, or what is the motivation. The paper refers to regularisation, but how (or why) do you regularise with Wasserstein?
- Earlier our data was u_j|D. Now in eq 8 it is instead tilde-u_obs. Can you please use consistent and harmonized notation throughout the paper?
- phi is undefined. Not sure where do you get the u in u_phi. Surely you don’t know the true underlying u (also, wasn’t the point that there is a distribution of true u’s, so which u are we talking about…).
- The presentation of eqs 8-10 is a mess, and I can’t follow. You can’t just add the |_D in eq 9, and I can’t follow where eq 10 is coming from. I also don’t understand the motivation of this entire thing, or what are we trying to do.

---

> ### Comment · Reviewer_z1Tq · 2024-05-19
>
> - Please don’t use ^t for both flow layers and sgld samples.
> - I can’t follow the SGLD sampling stuff. I’m not sure what is bar-u, or how we get a0. I’m not sure why do we want to change a^0 to a^t in the first place: I thought the goal was to keep “a” as simple GP, and only modify the flow part \theta. I’m confused what is the motivation and goal. It would be nice to have an illustration

---

> ### Author Response · Authors · 2024-05-21
> **Official Comment by Authors Part I**
>
> We appreciate the reviewer’s valuable comments. In the following, we address the comments and explain changes made in the main text. These changes in the main text are colored in blue in the revised draft.
>
> **Q1. Sec 3 UFR paragraph refers U to data space, but U is a function space. Usually data is vector-valued observations, not functions [unless eg. in audio]. It would be good to clarify the nature of “data” in this paper.**
>
> As the reviewer mentioned, in audio, data is a function in time and data space is a function space for which we have access to it at various sampling rates or observation points in time. In the study of “ML on function space” and “operator learning”, data points are often functions. For example, seismograms (which can be thought of as audio heard from Earth), or fluid dynamics flow which are functions in time and space.
> The current work is in the category of Machine Learning on function spaces, and data points are functions for which we have observation of them at some sampling rate and resolution. The sampling rate and observation points are often determined by the number of sensors used to record data, or by mesh sizes in computational simulations.
>
> **Q2. In sec 3 UFR paragraph you observe data at D_j, but evaluate them at D_dagger. Similarly you observe u_j, but consider density of u. I don’t really know what is the dagger stuff, or what is u (where does it come from?). Is u some kind of test point? I don’t really follow.**
>
> The setting is similar to the GP regression setting. In GP, we are given the function value on D_j points, i.e., u|D_j, and we are often interested in the function values at new points, which are the new points on D_dagger, i.e., u|D_dagger.
>
> **Q3. I don’t even understand the setting: I thought we were supposed to maximize the probability of the observed functions u_j, not some other mysterious function u.**
>
> Opflow has two steps. The first step of the OpFlow paradigm is the learning of prior on function spaces. This step is concerned with training a neural operator that maximizes the data likelihood p(u). After learning the prior, we have the UFR setting, which is the function regression part, in which we are concerned with the value of the function on new points p(u given u|D_dagger). These two steps are similar to how GP is used in practice. In GP, experts often look at holdout data and domain knowledge, and tune the GP parameters, e.g., kernel on it. This step is on par with tuning the prior. After the GP parameters are set, at the inference time, the expert applies GP regression methods to understand the value of the inference function at new points, i.e., p(u given u|D_dagger)
>
> **Q4. I also don’t understand what is the “p” in the max. Is this the true “p”, or some parametric “p”? I don’t really understand the problem at all. Are we trying to find some true underlying u; or maximize true probability of observations; or maximize some parametric learnt probability of observations (or true function); or maybe you want to learn a measure instead (ie. some set function \mu… from measure theory?).**
>
> To answer this question, let's see what is p in GP. In GP regression, we are given function value at D_dagger points, then we are asked what is posterior. We think that, this reviewer’s comment might have originated from an unfortunate typo we made in the draft when introducing UFR and the max.
> At the first phase of the problem, we are interested in learning the prior on the data function space. Which is based on learning a model that maximizes p(u) on the dataset.  During the second phase, we are interested in the posterior distribution p(u given u|D_dagger). Our approach is concerned with learning both likelihood p(u) as well is the posterior p(u given u|D_dagger). In the case of GP, the p(u) is Gaussian, and p(u given u|D_dagger) becomes Gaussian by construction.
>
> **Q5. Also, the optimisation task of max_u p(u) is clearly pathological, and can be optimised by having a Dirac function on any arbitrary u \in U.**
>
> Please refer to the comment above. We had a typo in the first equation in the paper. We are after maximizing p(u given u|D_dagger) for the regression task. In the URF, we care about the MAP estimate as well as sampling from the posterior distribution.
>
> **Q6. Even if the argmax refers to the u_j, you could still have a pathological solution by just overfitting to u_j. I don’t see anything in the formulation that would ensure generalisation (or even discuss it). The “posterior distribution” is undefined, so I’m not sure what it means.**
>
> Please refer to the comment above. Sorry again for the confusion. We had a typo, we are after maximizing p(u given u|D_dagger) for the regression task.
>
> **Q7. In GPR we do not try to maximize likelihood of data. Instead, we would find the posterior, and try to learn the kernel also.**
>
> Please refer to the comment above. We had a typo, we are after maximizing p(u given u|D_dagger) for the regression task.

---

> > ### Author Response · Authors · 2024-05-21
> > **Official Comment by Authors Part II**
> >
> > **Q8. It’s also vague what is the relationship between p and P_U. It’s also vague what is the relationship between p in the first paragraph of sec 3, and the p in the argmax. Are they the same p? I don’t think so, since the argmax-p is learnt, so surely it should contain some parameters.**
> >
> > Yes, they are the same p. In GRF, the prior p(u) is given. However, in this work, we need to first learn the prior p(u). In this paper, p(u) is the data prior coming from P_U measure. p(u|D) is the likelihood on D collocation points. And p(u | u|D) is p(u given u|D), that is the posterior, similarly, p(u|D’ | u|D) is the posterior on D’ collocation points. We elaborate on this notation in the Section 3.
> >
> > **Q9. The “invertible operators” paragraph repeats math. This gives the impression that the two paragraphs in sec 3 are two separate stories. Are they? It would be good to harmonize the material in the paper.**
> >
> > They are separate things. The first paragraph mathematically defines URF. This definition is independent of invertible operators. The second paragraph is mainly concerned with the generic setting of invertible operators and how they can map two measures.
> >
> > **Q10. “a \in A is drawn… and consequently a_j is a…”. Do we sample a or a_j? Where did the a_j come from?**
> >
> > Sorry for the typo, it’s “a_j \in A is drawn … “ and “a_j” is a function drawn from its own measure in an abstract sense, and we only query its values on D_A points, i.e., a_j|D_A.
> >
> > **Q11. How can you have an invertible mapping if A and U have different domains? Is this necessary?**
> >
> > It can be achieved e.g. first apply a homography transform of  D_A to D_U and then we have the full stack invertible operator between similar domains. Generally, this choice depends on the application. One can think of a setting where D_U has complex geometry, e.g., in material science function are defined on deformed space, and the domain expert prefers designing D_A as cube since the codes that sample on cube (using DCT or other techniques) are much faster than those that can sample on complex geometries
> >
> > **Q12. What does “small amount of GP noise” mean? What is “GP noise”? Why do you need to add extra noise, or why does it ensure continuity?**
> >
> > This is a GP draw with very small amplitude. The idea behind this step is the same as the typical approaches in normalizing flow and diffusion models where small Gaussian perturbation is added to the data to ensure well-definedness. If the data lives in a lower dimensional manifold, then we can not find an invertible map from GP to this data distribution, and back. Adding a small GP noise to the data resolves this issue. This is also the case for diffusion models on function spaces.
> >
> > **Q13. In eq 1 the P_U is unknown. How do you sample from it? You do observe u_j|D, but this is very different from obtaining entire u_j, which is what eq 1 calls for.**
> >
> > In Equation 1, \( P_U \) represents the probability measure from which the training data are sampled, assumed to be i.i.d. samples of \( \{u_j\}_{j=1}^m \sim P_U \). This sampling assumption aligns with conventional machine learning practices where learning is constrained by the extent and nature of the available data. As noted, obtaining an entirely accurate model would require an infinite number of data points covering all possible resolutions, which is practically not feasible. Consequently, our method inherently includes both generalization and approximation errors.
> >
> > Equation 1 outlines the training approach that allows the OpFlow model to handle datasets with varied discretizations for each data point, underscoring its discretization-agnostic capability. We discuss this property in detail in Appendix A.2 and further demonstrate in Appendix A.7 that OpFlow, even if trained on data from a single resolution, can  be effectively evaluated on higher resolution through zero-shot super-resolution experiments. This flexibility contrasts sharply with classical normalizing flow models like RealNVP or Glow, which require the same resolution across training and evaluation
> >
> > **Q14. I dont’ understand how the domain points evolve in eq 1, or why is that a valid thing to do. It would be good to have an illustration of the functional flow, and how it works. I don’t really understand this yet intuitively.**
> >
> > Thanks for the comment. For the sake of simplicity, we considered that all points locations on all the layers are identical to D in our experiments. Regarding equation 1, when fixing the resolution, the equation reduces to standard normalizing flow (identical to the work on normalizing flow). This equation is the Jacobian transform of maps between random variables. There are not many special steps beyond basic random variable derivations here. Could you please elaborate which part is concerning?

---

> > > ### Author Response · Authors · 2024-05-21
> > > **Official Comment by Authors Part III**
> > >
> > > **Q15. Eq 1 is still just appearing in this paper, and there is no derivation or construction of this. This is the main novelty of the paper, and it needs a strong exposition and derivation. I’m not convinced that this is a valid mathematical construct, and I suspect that the authors are casually assuming (or hoping) that standard normalizing flow can be written like this.**
> > >
> > > Equation 1 introduces the training approach for the OpFlow model, where each data point in the dataset can have its own discretization. This flexibility distinguishes OpFlow from traditional normalizing flow models like RealNVP or Glow, which require the same resolution across training and evaluation. In Appendix A.2, we elaborate why the training objective of OpFlow is valid and why the OpFlow learns the probability measures of the data. Moreover, experiments in Appendix A.7 demonstrate that OpFlow can be effectively evaluated at higher resolutions not seen in the training dataset.
> > >
> > >  The intuitive explanations are as follows: (1) The Neural Operator framework, which OpFlow leverages, is designed to learn mappings between infinite-dimensional function spaces through the point-evaluation of functions. During training, the integral kernel operator's weights are optimized, which involves Riemann summations. (2) The eq1 can take place for any discretization, thus it’s a valid objective in operator learning (Neural operator, Kovachki et al, 2021), the further introduced generalized W2 regularization also holds for any discretization. (3) As pointed above, the true invertible operator can only be trained when the number of data points is infinity and covers all resolutions from low to infinity. Since neither of these two ever would be achieved, our method will have both generalization error and approximation error. However, such error converges when you evaluate the model on finer discretizations which are not available in the training dataset. This statement is supported by the theory of Neural Operators  (Kovachki et al, 2021) and our zero-shot super-resolution experiment in Appendix A.7.
> > >
> > > **Q16. Last sentence of eq 1: I’m not sure I agree with this. Can you proof or show this? Why does the eq 1 “measure” distance?**
> > >
> > > We revised corresponding text, the reason why eq 1 is a valid objective is explained above.
> > >
> > > **Q17. The whole point of the paper is to do mappings between functions. Yet, it seems that you immediately discretise back into standard grids of images. Doesn’t this then just reduce to a standard normalizing flow? What extra does the OpFlow bring us if we discretise anyways?**
> > >
> > > In the area of machine learning on function spaces, we deal with functions at various discretization and proposed methods should be able to learn and operate on such discretization and output function at any required discretization. Our approach in this paper would only coincide with the development of standard normalizing flow on images if the discretization used was a fixed grid across all the data points, as well as inference time. Please also note that stand normalizing flow does not intersect with GP (data on function), while our method subsumes GP as a special case. We showed that once OpFlow is trained, we can query new values at any resolution (zeor-shot super-resolution experiments in Appendix A.7) and elaborate the resolution-agnostic property of our model in Appendix A.2. Standard normalizing flow, like RealNVP or Glow, only enables you to train the model using one resolution and evaluate the model on the same resolution. That is why people need to design and train separate normalizing flow models from scratch for super-resolution tasks ( SRFlow, Lugmayr et al 2020).

---

> > > > ### Author Response · Authors · 2024-05-21
> > > > **Official Comment by Authors Part IV**
> > > >
> > > > **Q18, I don’t understand where does GP_1, GP_2, K_1, K_2 come from. What are these? I struggle to follow the Wasserstein story. I struggle to understand what are we trying to achieve, or what is the motivation. The paper refers to regularisation, but how (or why) do you regularise with Wasserstein?**
> > > >
> > > > As explained in the paper, P_A is chosen to be GP_1(m_1, K_1), where m_1, K_1 are the mean function and covariance operator. This is to say we want to build the bijective mapping between data function space and GP_1(m_1, k_1) with OpFlow. GP_1(m_1, K_1) is defined by the user. GP_2(m_2, K_2) is the GP fit to the push-forward measure F_{\theta}#P_U. In the ideal case if OpFlow learns the bijective mapping between P_U and P_A, the GP_2(m_2, k_2) will exactly match the GP_1(m_1, k_2), where the 2-Wasserstein distance between GP_1 and GP_2 is 0. Since the 2-Wasserstein distance between two Gaussian Processes is always non-negative, then we can take it as a regularization. The motivation for using the 2-Wasserstein as the regularization is because we experienced a hard time training the OpFlow model using only the loss in equation 1. We hypothesized that might be due to architecture expressiveness and typical hardness of training invertible deep learning models. Given the prior success of the GANO (Rahman et al, 2022) model, we attempted to train our architecture in the GANO framework. This attempt showed that GANO loss (infinite-dimensional Wasserstein loss) can considerably help our model training procedure
> > > >
> > > > **Q19. Earlier our data was u_j|D. Now in eq 8 it is instead tilde-u_obs. Can you please use consistent and harmonized notation throughout the paper?**
> > > >
> > > > As pointed in the text right above eq 8, tilde-u_obs refers to the noise observation, and u_obs refers to the ground truth without the additive noise. These notations are used to distinguish from the notation of the training data u_j|D.
> > > >
> > > > **Q20. phi is undefined. Not sure where do you get the u in u_phi. Surely you don’t know the true underlying u (also, wasn’t the point that there is a distribution of true u’s, so which u are we talking about…).**
> > > >
> > > > As pointed in the text right above eq 8. We refer to the inferred function’s values as u_phi to distinguish it from the training data u, because the regression domain of u_phi can be different from the discretized domain (collection of positions) of u in the training dataset. The equation 8-10 is used to describe the posterior distribution of u_phi given some observations.
> > > >
> > > > **Q21. The presentation of eqs 8-10 is a mess, and I can’t follow. You can’t just add the |_D in eq 9, and I can’t follow where eq 10 is coming from. I also don’t understand the motivation of this entire thing, or what are we trying to do.**
> > > >
> > > > Equation 8 introduces the second critical phase of OpFlow, focusing on regression. Here, u_j represent training data used to learn the prior, and during inference, we are presented with \tilde{u}_{\text{obs}}, a noisy observation of new functional data.
> > > >
> > > > To detail equations 8-10: \log p_{\theta}(\tilde{u}\_{\text{obs}} | u_\phi) starts by considering \tilde{u}\_{\text{obs}}. The observation defined over domain D with u\_\phi as the inferred function over domain D'. Importantly, since D \subset D', it follows that \log p_{\theta}(\tilde{u}_{\text{obs}} | D \text{ given } u_\phi | D') = \log p_{\theta}(\tilde{u}_{\text{obs}} | D \text{ given } u_\phi | D). This derivation forms the basis of Equation 9 from Equation 8.
> > > >
> > > > Transitioning to Equation 10, we consider the assumption that the additive white noise, denoted as \epsilon, follows a normal distribution N(0, \sigma^2). This assumption provides a closed-form expression for \log p_{\theta}(\tilde{u}_{\text{obs}} | D \text{ given } u\_\phi | D), which describes the distribution of the white noise as a multivariate Gaussian. This approach aligns with the noise level used in GPR, thereby clarifying the progression from Equation 9 to Equation 10 and underscoring its theoretical grounding.

---

> > > > > ### Author Response · Authors · 2024-05-21
> > > > > **Official Comment by Authors Part V**
> > > > >
> > > > > **Q22. Please don’t use ^t for both flow layers and sgld samples.**
> > > > >
> > > > > Thanks for the suggestion, we addressed this in the main text.
> > > > >
> > > > > **Q23. I can’t follow the SGLD sampling stuff. I’m not sure what is bar-u, or how we get a0. I’m not sure why do we want to change a^0 to a^t in the first place: I thought the goal was to keep “a” as simple GP, and only modify the flow part \theta. I’m confused what is the motivation and goal. It would be nice to have an illustration**
> > > > >
> > > > > We thank the reviewer for this valuable comment. We assume the reviewer is referring to Algorithm 2, please correct us if we misunderstood the reviewer’s point.
> > > > >
> > > > > \overline{u}\_{\phi} is defined as the Maximum A Posteriori (MAP) estimate given the noisy observations.  a^0 is the push-forward of \overline{u}\_{\phi} through the OpFlow F_{\theta}, denoted as a^0 = F_{\theta}(\overline{u}\_{\phi}). This sets a^0 as the starting point for the SGLD procedure.  a^0  serves as the initialization point for the iterative process, where t denotes the iteration index starting from 0.
> > > > >
> > > > > During regression tasks, the trained flow is frozen. Through Equations 8-10) combined with SGLD, we can directly sample the target function u_{\phi} from the posterior distribution. The ability to directly sample u_{\phi} from \mathcal{U} is feasible because our trained prior is fully differentiable. However, our experimental findings suggest that sampling within the \mathcal{A} space (Gaussian process space) and then mapping to \mathcal{U} yields better performance compared to direct sampling in data function spaces \mathcal{U}. The motivation is to efficiently sample from the true posterior distribution. By initializing SGLD at the MAP estimate, we combine the robustness of starting at a high-probability point with the flexibility to explore the entire posterior. This method reduces the number of iterations needed to achieve convergence while ensuring thorough exploration of the parameter space to accurately represent the posterior's complexity.
> > > > >
> > > > > For a visual representation, please refer to Figure 1 in the manuscript, which illustrates the mapping and sampling process described above. We hope this response provides the necessary clarifications regarding the use and motivation of our algorithm. We are open to further discussions if there are additional questions or if any points remain unclear.

---

### Comment · Action_Editor_Q5Jc · 2024-07-19
**Additional changes needed**

Dear authors

Thanks for the reply to the reviewer's comments. I'm mostly happy about the way you have addressed the discussion.

However, after reading the discussion and the paper, I believe there are a few points that need to be undertaken before the paper can be accepted:

1. There needs to be an experimental comparison against Deep GPs and Maroñas et al (2021). It is difficult for the audience to see how the theory claims in the paper translate into specific benefits in terms of methods that already exist. The potential benefits of the proposed approach should be exemplified in comparison to methods that already do similar jobs. Otherwise, it is likely the significance of the work is lower.

2.  You have added this paragraph to the paper after discussing it with the reviewers: "Regression with OpFlow addresses the limitations of existing Gaussian process models discussed above, and offers two key improvements over the Transformed GP framework (Maroñas et al., 2021). To be specific, regression with OpFlow enables: (i) Non-diagonal transformations while ensuring that the output of each layer of OpFlow is a valid stochastic process, which allows modelling arbitrary stochastic processes with complex correlation structures. (ii) Robust and accurate uncertainty quantification via drawing posterior samples of the Gaussian process with SGLD and subsequently mapping them into the data function space, thereby obviating the need for variational inference."

Here you need to further substantiate your claims:
i) "Non-diagonal transformations while ensuring that output of each layer of OpFlow is a valid stochastic process, which allows modeling arbitrary stochastic process with complex correlation structures."
a) Show (empirically) that by providing non-diagonal transformations, the output of each later of OpFlow is a valid stochastic process, contrary to what happens with purely diagonal transformations, a la Maroña et al (2021).
b) This sentence "which allows modelling arbitrary stochastic process with complex correlation structures" is rather problematic. I don't see any theorems in the paper where you prove this is the case. What do you mean exactly by "arbitrary stochastic process"?

ii)  "Robust and accurate uncertainty quantification via drawing posterior samples of the Gaussian process with SGLD and subsequently mapping them into the data function space, thereby obviating the need for variational inference."

Again, you need to further substantiate your claims.
a) "Robust and accurate uncertainty quantification" What do you mean by robust here? Is there a performance metric of robustness that you can show empirically is better than using other approaches? Or can you please direct the reader to the proper experiments where this is shown? Likewise for "accurate".
b) "thereby obviating the need for variational inference" There is an implication here that variational inference is good to avoid. Can you further discuss this point and explain why your method excels at any particular metric that the variational inference does not get right?

Thanks

---

> ### Author Response · Authors · 2024-07-24
> **Official Comment by Authors Part I**
>
> Dear Mauricio,
>
> Thank you very much for your constructive comments, we appreciate your insights and the chance to enhance the clarity and impact of our work. We have carefully considered your suggestions for additional comparison and substantiations regarding our claims, In the following, we address the comments and explain changes made in the main text. These changes in the main text are colored in blue in the revised draft.
>
> **Q1. There needs to be an experimental comparison against Deep GPs and Maroñas et al (2021). It is difficult for the audience to see how the theory claims in the paper translate into specific benefits in terms of methods that already exist. The potential benefits of the proposed approach should be exemplified in comparison to methods that already do similar jobs. Otherwise, it is likely the significance of the work is lower.**
>
> This is a great suggestion. We show the additional experimental comparison in Appendix A.4, which demonstrates the superior regression performance of OpFlow. Our results indicate that OpFlow provides a more accurate predicted mean, standard deviation, and more realistic posterior samples compared to Deep GP
>
> Furthermore, it is important to note that the regression framework of OpFlow slightly differs from that of traditional Deep GP models. In typical Deep GP or Transformed GP frameworks (Maroñas et al.), the model is fitted to the entire dataset, with each data point treated as a point evaluation of a function. For instance, when using a dataset of rainfall measurements spanning ten years, a Deep GP or Transformed GP would directly fit all the data, thereby enabling predictions at any point within this period. In contrast, the regression approach with OpFlow can be likened to image inpainting or reconstruction from pixels with generative models in a Bayesian manner (Marinescu et al., 2021). Within the OpFlow framework, each data point in the training set represents a distinct function, potentially with its unique discretization. During training, the goal of OpFlow is to learn the bijective mapping between a Gaussian process and the data function space. Once trained, OpFlow is frozen and serves as the prior. When provided with new observations (i.e., point evaluations of an unknown function drawn from the data function measure), the task becomes reconstructing the entire function, thus aligning OpFlow closely with the conventional settings of standard GP regression. This distinction is highlighted in the revised draft.
>
> We also attempted to incorporate a comparison with Transformed GP (Maroñas et al., 2021); however, despite the high-quality code provided by the authors, no tutorials were available that demonstrate how to use their model similarly to traditional GP regression. Efforts to modify their code were hampered by extensive wrappings functions, which made alterations challenging.
>
>
> Reference :
>
> Razvan V. Marinescu, Daniel Moyer, and Polina Golland. Bayesian Image Reconstruction using Deep Generative Models, December 2021. URL http://arxiv.org/abs/2012.04567. arXiv:2012.04567 [cs, eess, stat]

---

> > ### Author Response · Authors · 2024-07-24
> > **Official Comment by Authors Part II**
> >
> > **Q2. Here you need to further substantiate your claims: i) "Non-diagonal transformations while ensuring that output of each layer of OpFlow is a valid stochastic process, which allows modeling arbitrary stochastic process with complex correlation structures." a) Show (empirically) that by providing non-diagonal transformations, the output of each later of OpFlow is a valid stochastic process, contrary to what happens with purely diagonal transformations, a la Maroña et al (2021). b) This sentence "which allows modelling arbitrary stochastic process with complex correlation structures" is rather problematic. I don't see any theorems in the paper where you prove this is the case. What do you mean exactly by "arbitrary stochastic process"?**
> >
> > This claim was initially posited in section 3.4 of Marona et al. (2021), where the authors noted that the normalizing flow used in Transformed GP could be extended to non-diagonal transformations, such as RealNVP (Dinh et al., 2017), to model arbitrary copulas. However, they also highlighted the need to demonstrate that such transformations should induce a valid stochastic process from each layer of the normalizing flow. Building upon this concept, OpFlow generalizes RealNVP to function space, thus enabling non-diagonal transformations for any finite collection of points while still inducing a valid stochastic process for each layer of the model. An illustrative example of non-diagonal transformation using normalizing flow is depicted in Figure 1C of Ricky et al. (2019).
> >
> > OpFlow extends the capabilities of the marginal flow utilized in Marona et al. (2021) by removing restrictions on the underlying form of the data function measure during training. This freedom allows OpFlow to learn the mapping between a GP with an underlying unknown arbitrary stochastic process through non-diagonal transformation. This claim is also supported by the fact that we don’t have assumptions about the type of stochastic process that governs the data function measure which generates the training data. Such an advancement demonstrates that OpFlow is capable of learning a more general stochastic process compared to the marginal flow approach used by Marona et al. (2021).
> >
> > Reference:
> >
> > Ricky T. Q. Chen, Jens Behrmann, David K. Duvenaud, and J¨orn-Henrik Jacobsen. Residual flows for invertible generative modeling. In Advances in Neural Information Processing Systems, 2019
> >
> > **Q3. Again, you need to further substantiate your claims. a) "Robust and accurate uncertainty quantification" What do you mean by robust here? Is there a performance metric of robustness that you can show empirically is better than using other approaches? Or can you please direct the reader to the proper experiments where this is shown? Likewise for "accurate". b) "thereby obviating the need for variational inference" There is an implication here that variational inference is good to avoid. Can you further discuss this point and explain why your method excels at any particular metric that the variational inference does not get right?**
> >
> > This is a great comment. Variational inference and Stochastic Gradient Langevin Dynamics (SGLD) are two prominent methods employed in Bayesian inference. Generally, SGLD is known to provide more accurate (unbiased in the limit) samples compared to variational inference, the latter introduces bias because the true posterior may not lie within the chosen family of distributions. However, SGLD requires meticulous tuning of parameters such as step size and temperature, etc, and its performance is highly contingent on the complexity of the problem. When dealing with functions with high curvature, SGLD may encounter issues with trapping in local minima and convergence. Nonetheless, when applied within a Gaussian Process space, these issues are circumvented due to the inherent smoothness of Gaussian Processes.
> >
> > With OpFlow, the complexity of the data function measure is less concerning. Thanks to its bijective structure, we can implement SGLD within a selected Gaussian Process space. Subsequently, OpFlow facilitates the transformation of posterior samples from the Gaussian space back to the data function space, thus enabling robust SGLD sampling that is less sensitive to the regression parameters. In Appendix A.3, we empirically demonstrate that even across various regression problems, the same regression parameters can be used. All posterior samples closely approximate the true posterior samples, underscoring the robustness of combining SGLD with OpFlow. Here, our focus is solely on the quality of the posterior samples as discussed in the main text. While SGLD can yield better posterior samples, it requires significantly more time for sampling compared to variational inference.

---

> > > ### Comment · Action_Editor_Q5Jc · 2024-07-26
> > > **illustrative example of Ricky et al. (2019)**
> > >
> > > Regarding my previous comment on your claims on "Non-diagonal transformations" and "which allows modelling arbitrary stochastic process". You're now introducing new claims and, I don't think you answered my question. Again you say: "OpFlow generalizes RealNVP to function space, thus enabling non-diagonal transformations for any finite collection of points while still inducing a valid stochastic process for each layer of the model." How do we know each layer of the model is inducing a valid stochastic process? Again, valid in what sense? There is a strict definition of what can and can't be a stochastic process using the Kolmogorov extension theorem. Is this what you're referring to when you say valid? And if so, how is that true for your case?
> > >
> > > Again, on my comment on robustness, you didn't answer my question. "What do you mean by robust here? Is there a performance metric of robustness that you can show empirically is better than using other approaches?" Again, you're making all these claims on what variational inference is doing, but how come your model is not doing something similar (e.g. bias)? I don't see any discussion related to the limitations of your approach.

---

> > ### Comment · Action_Editor_Q5Jc · 2024-07-26
> > **The experimental setting for Deep GPs in Appendix A.4 is outdated**
> >
> > Thanks for pointing me to Appendix A.4. You are not comparing against the state-of-the-art models in Deep GPs. No one uses the original formulation of Damioanou and Lawrence from 2013 anymore since it's well known it has quite a few pathologies. The state-of-the-art deep GP version that is still used as a baseline is the one by Salimbeni and Deisenroth (2017), "Doubly stochastic variational inference for deep Gaussian processes". Comparisons against this model are the minimum. As you can see, Maroñas et al (2021) have quite a few empirical comparisons against this version of deep GPs. Furthermore, it seems most of your experiments are for 1-D and 2-D datasets, have you discussed the performance of your model for higher-dimensional datasets? And have you done any experiments in that regard?

---

> > > ### Comment · Action_Editor_Q5Jc · 2024-07-26
> > > **Performance metrics used in table 3**
> > >
> > > Can you define what performance metrics are RMSE in the mean and RMSE in the std? These are not metrics used in the GP literature. Can you please provide metrics more common in the GP literature such as the Standardised MSE and the mean standardised log loss (MSLL)?

---

> > > > ### Author Response · Authors · 2024-08-01
> > > > **Official Comment by Authors Part I**
> > > >
> > > > Dear Mauricio,
> > > >
> > > > We very much appreciate your comments and please accept our apologies for the earlier reply. Last week, some of us were overwhelmed with ICML and our star junior student took the lead and liberty of replying to your comments. Thanks for kindly pointing out the limitations of the response and thanks for helping us to raise independent researchers for the future.
> > > >
> > > > In the following please find the replies to your comments.
> > > >
> > > > **Q1. Thanks for pointing me to Appendix A.4. You are not comparing against the state-of-the-art models in Deep GPs. No one uses the original formulation of Damioanou and Lawrence from 2013 anymore since it's well known it has quite a few pathologies. The state-of-the-art deep GP version that is still used as a baseline is the one by Salimbeni and Deisenroth (2017), "Doubly stochastic variational inference for deep Gaussian processes". Comparisons against this model are the minimum. As you can see, Maroñas et al (2021) have quite a few empirical comparisons against this version of deep GPs. Furthermore, it seems most of your experiments are for 1-D and 2-D datasets, have you discussed the performance of your model for higher-dimensional datasets? And have you done any experiments in that regard? Can you define what performance metrics are RMSE in the mean and RMSE in the std? These are not metrics used in the GP literature. Can you please provide metrics more common in the GP literature such as the Standardised MSE and the mean standardised log loss (MSLL)?**
> > > >
> > > > Thank you for your valuable feedback.  We apologize for the oversight in our baseline selection and appreciate your guidance in aligning our comparisons with the current standards in the field. In the following, we address the comments and explain changes made in the main text. These changes in the main text are colored in blue in the revised draft.
> > > > In the revised Appendix A.4 of our manuscript, we have now included comparisons against the state-of-the-art Deep GP modes (Salimbeni and Deisenroth (2017)), and Deep Sigma Point Processes (Jankowiak et al. (2020)). We utilized the official implementations from GPyTorch (Gardner et al., 2021) to ensure repeatable results, and report best performances of baseline models for fair comparison.
> > > >
> > > > Regarding the performance metrics, we have replaced the previously used RMSE metrics with the more standard GP literature metrics of Standardized Mean Squared Error (SMSE) and Mean Standardized Log Loss (MSLL). In Appendix A.4, we show that OpFlow generates realistic posterior samples and achieves lower SMSE and MSLL values compared to the baseline models. Currently, our research primarily focuses on 1D and 2D datasets, employing operator learning techniques such as those developed for the Fourier Neural Operator (FNO) by Li et al. (2021). Learning operators for functions defined on high-dimensional domains remains challenging and is an area that has not been thoroughly developed, neither from computation standpoint nor on the datasets side (Kovachki et al. (2023)). It is also custom to use 1D and 2D functions to develop early steps of developments on ML on function spaces, while in parallel the field develops datasets.
> > > >
> > > > We hope the updated experiments address the concerns raised in comparison with other Deep GP models.
> > > >
> > > >
> > > > References:
> > > >
> > > > Hugh Salimbeni and Marc Deisenroth. Doubly Stochastic Variational Inference for Deep Gaussian Processes, November 2017. URL http://arxiv.org/abs/1705.08933. arXiv:1705.08933 [stat].
> > > >
> > > > Martin Jankowiak, Geoff Pleiss, and Jacob R. Gardner. Deep Sigma Point Processes, December 2020. URL. http://arxiv.org/abs/2002.09112. arXiv:2002.09112 [cs, stat].
> > > >
> > > > Jacob R. Gardner, Geoff Pleiss, David Bindel, Kilian Q. Weinberger, and Andrew Gordon Wilson. GPyTorch: Blackbox Matrix-Matrix Gaussian Process Inference with GPU Acceleration, June 2021. URL http: //arxiv.org/abs/1809.11165. arXiv:1809.11165 [cs, stat].
> > > >
> > > > Zongyi Li, Nikola Kovachki, Kamyar Azizzadenesheli, Burigede Liu, Kaushik Bhattacharya, Andrew Stuart, and Anima Anandkumar. Fourier Neural Operator for Parametric Partial Differential Equations, May 2021. URL http://arxiv.org/abs/2010.08895. arXiv:2010.08895 [cs, math]
> > > >
> > > > Nikola Kovachki, Zongyi Li, Burigede Liu, Kamyar Azizzadenesheli, Kaushik Bhattacharya, Andrew Stuart, and Anima Anandkumar. Neural Operator: Learning Maps Between Function Spaces, April 2023. URL http://arxiv.org/abs/2108.08481. arXiv:2108.08481 [cs, math].

---

> > > > > ### Author Response · Authors · 2024-08-01
> > > > > **Official Comment by Authors Part II**
> > > > >
> > > > > **Q2. Regarding my previous comment on your claims on "Non-diagonal transformations" and "which allows modelling arbitrary stochastic process". You're now introducing new claims and, I don't think you answered my question. Again you say: "OpFlow generalizes RealNVP to function space, thus enabling non-diagonal transformations for any finite collection of points while still inducing a valid stochastic process for each layer of the model." How do we know each layer of the model is inducing a valid stochastic process? Again, valid in what sense? There is a strict definition of what can and can't be a stochastic process using the Kolmogorov extension theorem. Is this what you're referring to when you say valid? And if so, how is that true for your case?**
> > > > >
> > > > > We apologize for any confusion caused by the imprecise language used in our initial manuscript. We have carefully reviewed the text and revised the claim that OpFlow models 'arbitrary stochastic processes' to a more accurate description: 'OpFlow is able to model a more general stochastic process compared to the marginal flow used in Maroñas et al. (2021).' In Appendix A.5 of the revised manuscript, we have formally defined the problem setting and verified that valid stochastic processes can be induced from OpFlow via the Kolmogorov extension theorem.
> > > > >
> > > > > In finite dimensional spaces, normalizing flow is provided to have universal representation for any data distribution under some mild conditions ((Kong & Chaudhuri, 2020),  section 2.2 of (Papamakarios et al., 2021).  Following the fact that universal approximation of normalizing flow, we expect such universal approximation also be generalizable to OpFlow, similar to its finite-dimensional case. However, providing a rigorous proof of universal approximation of OpFlow remains beyond the scope of this paper and we leave it for future work. In this paper, we focus on demonstrating: (1) Valid stochastic processes can be induced from OpFlow; (2) OpFlow can model a more general stochastic process compared to the baseline (marginal flow in Marons et al, 2021 which is a special case of OpFlow) . Here is the intuitive explanation: The marginal flow is based on pointwise transformation of GP values, ergo, a pointwise operator. It induces valid stochastic processes and since it is limited to only pointwise transformation of GP values, it  produces transformations with diagonal Jacobians. In contrast, OpFlow acts as a more general bijective operator that results in transformations with triangular Jacobian matrix (non-diagonal) and jointly transforms the GP values. We formally define the problem setting and provide a proof of our statements via Kolmogorov extension theorem in Appendix A.5.
> > > > >
> > > > > Reference:
> > > > >
> > > > > Zhifeng Kong and Kamalika Chaudhuri. The Expressive Power of a Class of Normalizing Flow Models, May 2020. URL http://arxiv.org/abs/2006.00392. arXiv:2006.00392 [cs, stat]
> > > > >
> > > > > George Papamakarios, Eric Nalisnick, Danilo Jimenez Rezende, Shakir Mohamed, and Balaji Lakshminarayanan. Normalizing flows for probabilistic modeling and inference. J. Mach. Learn. Res., 22(1): 57:2617–57:2680, January 2021. ISSN 1532-4435

---

> ### Author Response · Authors · 2024-08-01
> **Official Comment by Authors Part III**
>
> **Q3. Again, on my comment on robustness, you didn't answer my question. "What do you mean by robust here? Is there a performance metric of robustness that you can show empirically is better than using other approaches?" Again, you're making all these claims on what variational inference is doing, but how come your model is not doing something similar (e.g. bias)? I don't see any discussion related to the limitations of your approach.**
>
> We apologize for not addressing your concerns regarding robustness more directly in our previous communications. To clarify, we have revised Section 2 of the manuscript to ensure the language clearly articulates our claims without ambiguity.
>
> In response to your question about robustness, we describe our regression framework as robust primarily because of its principled approach to capturing true Bayesian uncertainty. The bijective structure of OpFlow allows for the implementation of SGLD within a selected Gaussian Process space. We can then map the posterior samples back to the data function space, which enables us to capture true Bayesian uncertainty. We agree that robustness was not the wisest choice to describe the fact that our “formulation” endures no approximation. We change the term robust to a more semantically relevant term, “valid”.
>
> In contrast to our approach, variational inference relies on optimizing the Evidence Lower Bound (ELBO) and yields an approximate posterior with an approximation gap “in the formulation”, rather than the true posterior. Please note that both methods have approximation errors due to limited data, expressivity, and optimization challenges. However, here in the above statement, we are only concerned with the problem “formulations”.
>
> We acknowledge that our approach is not without limitations. Notably, it requires a significantly longer time for posterior sampling compared to variational inference. This extended sampling time can render our approach less practical for time-sensitive applications. We bring this point into the main text.

---

### Decision · Action_Editor_Q5Jc · 2024-09-23

**Recommendation:** Accept with minor revision

**Comment:**

The authors have made a great effort to address the comments of the reviewers and the editor. The paper has improved and now contains better-substantiated claims and further experimental results that support the claims. As a minor revision, the authors need to include the limitations of their work prominently, perhaps as part of the Conclusion. As a minimum, the authors need to say the current model only allows 1-D or 2D input dimensional datasets, as was confirmed by the authors in the discussion.

**Audience:**

There is indeed an audience for the models and results in this paper including those working on non-Gaussian stochastic processes.

**Claims And Evidence:**

After discussing with the authors, several claims have now been clarified, and they better reflect the qualities of the proposed model and the experimental results, in particular those related to a previous model by Maroñas et al. (2021).